# Microstructural Effects on Irradiation Creep of Reactor Core Materials

**DOI:** 10.3390/ma16062287

**Published:** 2023-03-13

**Authors:** Malcolm Griffiths

**Affiliations:** 1Department of Mechanical & Aerospace Engineering, Carleton University, Ottawa, ON K1S5B6, Canada; malcolmgriffiths@cunet.carleton.ca; Tel.: +1-613-585-3315; 2Department of Mechanical and Materials Engineering, Queen’s University, Kingston, ON K7L3N6, Canada; 3ANT International, 448 50 Tollered, Sweden

**Keywords:** zirconium alloys, austenitic stainless steel, irradiation, nuclear reactors, swelling, creep, microstructure, rate theory, dislocation slip, texture, line broadening

## Abstract

The processes that control irradiation creep are dependent on the temperature and the rate of production of freely migrating point defects, affecting both the microstructure and the mechanisms of mass transport. Because of the experimental difficulties in studying irradiation creep, many different hypothetical models have been developed that either favour a dislocation slip or a mass transport mechanism. Irradiation creep mechanisms and models that are dependent on the microstructure, which are either fully or partially mechanistic in nature, are described and discussed in terms of their ability to account for the in-reactor creep behaviour of various nuclear reactor core materials. A rate theory model for creep of Zr-2.5Nb pressure tubing in CANDU reactors incorporating the as-fabricated microstructure has been developed that gives good agreement with measurements for tubes manufactured by different fabrication routes having very different microstructures. One can therefore conclude that for Zr-alloys at temperatures < 300 °C and stresses < 150 MPa, diffusional mass transport is the dominant creep mechanism. The most important microstructural parameter controlling irradiation creep for these conditions is the grain structure. Austenitic alloys follow similar microstructural dependencies as Zr-alloys, but up to higher temperature and stress ranges. The exception is that dislocation slip is dominant in austenitic alloys at temperatures < 100 °C because there are few barriers to dislocation slip at these low temperatures, which is linked to the enhanced recombination of irradiation-induced point defects.

## 1. Introduction

Creep is the time-dependent deformation of a material in response to an applied stress. Outside of the reactor environment, creep is thermally activated, either through increased atomic vibration aiding dislocation motion or the elevation of the vacancy concentration that enables dislocation climb over barriers [1]. At sufficiently high temperatures and low stresses (below yield), the effect of an applied stress can affect the flow of vacancies and their net emission or absorption at dislocations and grain boundaries. Creep is then governed by diffusional mass transport [2,3].

At high stresses (>150 MPa), the out-reactor creep of engineering alloys is dominated by dislocation slip, irrespective of temperature [4]. At very high temperatures (≥600 °C) and low stresses (<150 MPa), out-reactor creep is dominated by mass transport (bulk diffusion) or other mechanisms involving grain boundary diffusion or sliding [4]. Although in this paper there may be some reference to out-reactor creep, which is sometimes called thermal creep to distinguish it from irradiation creep, the intent is to focus on irradiation creep that is in the temperature range of interest to fission reactors. For sodium-cooled fast reactors (SFRs), operating temperatures vary from 400 °C to 550 °C for the internal structures, and up to 650 °C for the fuel cladding. The internal structures of the light water and heavy water reactors (LWRs and HWRs) operate at temperatures between about 200 °C and 320 °C, although some parts can be hotter (more than 400 °C) because of localised nuclear heating.

The irradiation creep of engineering alloys in nuclear applications is a technologically important topic, whether one is considering: (i) the expansion/contraction of cladding tubes in both fast (liquid metal cooled) and thermal (heavy and light water-cooled) reactors; (ii) the expansion and elongation of pressure tubing in pressure tube reactors (CANDU and RBMK design); or (iii) the creep relaxation of tensioning bolts and spring components in all reactor types. The creep of core components made from Zr-based hexagonal-close-packed (HCP) alloys and Fe, Cr and Ni-based face-centred-cubic (FCC) austenitic alloys in power reactors is of particular interest.

Zr-alloys have sufficiently good mechanical and corrosion properties, while also having low thermal neutron capture cross-sections, to be used for pressure tubing and as fuel cladding in LWRs and HWRs. Fuel cladding in liquid metal reactors does not have the same limitations as thermal power reactors vis-à-vis the need to have low thermal neutron absorption, and they are typically made from austenitic stainless steels or other high strength, ductile materials such as austenitic Ni-alloys [5,6,7,8,9]. The interest in the irradiation creep response of austenitic steels and Ni-alloys is mainly limited to bolts and springs and the loss/relaxation of tension for pre-tensioned material. Irradiation affects the mechanical strength and creep response of core materials mainly because of microstructure evolution, but chemical and phase changes may also occur. Whereas the radiation-induced redistribution of different elements and transmutation effects may affect material properties [10,11], the potential effect on irradiation creep will not be discussed in this paper.

Microscopic cavities can have a profound effect on the irradiation creep properties because cavities (like other point defect clusters) are barriers to dislocation slip but, perhaps more importantly, because of their effect on diffusional mass transport [7,8,9,12,13]. The main difference between cavities and dislocation loops as features that impede dislocation slip/glide is that dislocation loops can be swept up by the gliding dislocations, thus softening the material in channels, but cavities cannot [9]. Such channeling is a phenomenon that is observed close to the ultimate tensile strength (UTS) of the material and is not a concern here because the stresses during normal operation where irradiation creep is important are not high enough to induce the mechanical yielding that results in channeling. Irradiation creep will thus be limited to those stresses below the yield stress (YS) that exist in reactor components during normal operation, typically <150 MPa for Zirconium alloys, but may be higher for austenitic alloy springs and bolts.

Irradiation creep is particularly important when considering: (a) the interaction of fuel with fuel cladding, (b) the limits to operation caused by the diametral expansion of pressure tubes in CANDU and RBMK reactors, and (c) the relaxation of spring components in reactor cores. To this end, irradiation creep is largely detrimental to reactor operation. However, stress relaxation by irradiation creep can be beneficial in cases where the reduction of high stresses at the crack tips helps to mitigate stress corrosion cracking (SCC) in the case of austenitic steels and Ni-alloys [14] or delayed hydride cracking (DHC) for Zr-alloys [15,16]. With respect to SCC, Garzarolli showed that, whereas higher creep rates are beneficial in blunting cracks, they do not account for the relative propensity for the SCC of austenitic alloys [17]. High stress irradiation creep at the crack tips is important for flaw disposition in Zr-alloy components that are prone to delayed hydride cracking.

The irradiation creep behaviour in this paper is primarily directed at the effect of the microstructure on the creep itself. One needs to consider the as-fabricated microstructure and the microstructure that evolves during irradiation, primarily in the form of point defect clusters (dislocation loops and cavities), although the recrystallisation and recovery of the original microstructure may need to be considered for higher temperature irradiation. This paper includes a review of the relevant results on the mechanisms of irradiation creep that can be linked directly with the microstructure. To illustrate how the as-fabricated microstructure, together with a knowledge of how the microstructure evolves during irradiation, can be used to predict creep, a rate-theory model is described and applied to Zr-2.5Nb pressure tubing. The model is general in its applicability to Zr-alloys and austenitic alloys when the conditions dictate, i.e., when the stress and temperature are low enough. Use will be made of the many different excellent reviews on irradiation creep to help maintain the focus on the microstructure dependence.

## 2. Irradiation Creep in Nuclear Reactor Cores

### 2.1. Empirical Models

Microstructurally based models of irradiation creep have been the subject of debate for many years. There are two basic models that can be applied—one that includes swelling (specific to austenitic alloys), and one that does not (specific to Zr-alloys or austenitic alloys at low doses prior to the onset of swelling).

Empirical models for austenitic stainless steels and Ni-alloys are less complicated than those for Zr-alloys because they are deemed isotropic. Whereas the separation of swelling and creep are problematic for austenitic alloys [5,6,13], the anisotropic irradiation response increases the complexity for Zr-alloys [18,19].

At low stresses (<150 MPa), austenitic stainless steel and Ni-alloy components exhibit irradiation creep that is enhanced by irradiation and is linear with stress [5,6]. In fast reactors at high temperatures (above 400 °C), when the swelling is significant, the steady-state irradiation creep can be described by a simple empirical law relating the effective strain rate, $\dot{\bar{\varepsilon }}$, to the effective stress, $\bar{\sigma }$ [20], based on work by Foster et al. [21],
(1)$$\dot{\bar{\varepsilon }}=\left(B_{0}+D\dot{S}\right)\bar{\sigma }$$
where $B_{0}$ is the steady state irradiation creep rate compliance in the absence of swelling in units of dpa^−1^.MPa^−1^, *D* is the irradiation creep-swelling compliance in units of MPa^−1^ and $\dot{S}$ is the linear swelling rate, which is independent of stress, in units of dpa^−1^. There are no separate thermal and irradiation creep components to $B_{0}$, and the term $D\dot{S}$ is dependent on diffusional mass transport.

Many irradiation creep models for Zr-alloys assume that the long-term, steady-state plastic deformation consists of separable, additive components from thermal creep, irradiation creep and irradiation growth (or swelling) [18,19]. The deformation behaviour can be described empirically by
(2)$$\begin{array}{cccccc} {\dot{\varepsilon }}_{d}^{\mathrm{total}} & ={\dot{\varepsilon }}_{d}^{\mathrm{thermal}\mathrm{creep}} & + & \;{\dot{\varepsilon }}_{d}^{\mathrm{irradiation}\mathrm{creep}} & + & \;{\dot{\varepsilon }}_{d}^{\mathrm{irradiation}\mathrm{growth}\mathrm{or}\mathrm{swelling}}\\ & f\left(M,\;\sigma ,T\right) & & f^{\prime }\left(M,\;\sigma ,\Phi ,T\right) & & f^{\prime \prime }\left(M,\;\Phi ,T\right)\; \end{array}$$
where *f*, *f*′ and *f*″ are functions of the material and operating conditions. The operating conditions are stress (σ), temperature (T), and neutron flux (Φ). The response to these external constraints is dependent on the material or microstructure (*M*), which is dependent on the fabrication and irradiation history. For slip-based models, *M* is determined by the texture and is thus a term that captures the anisotropic response of the material to the applied stress, assuming that the anisotropy of the polycrystal is a function of the anisotropic deformation of the single crystal. The single crystal deforms according to the differences in critical resolved shear stress (CRSS) for dislocation slip on different systems. The thermal creep applies to out-reactor creep (zero flux) of the irradiated material. The irradiation creep also applies to irradiated material but includes the strain because of the point defects produced by the irradiation. The thermal creep and irradiation creep are governed by the same mechanism of dislocation slip [18], however they are treated separately because the thermal creep is experimentally found to be essentially isotropic, whereas the irradiation creep term is not. The third part is the strain observed in a neutron flux independent of stress, which is a shape change (irradiation growth) in the case of Zr-alloys but may also include a volume expansion if swelling is occurring. Whereas the rationale for separating the creep into three parts is based on simplicity, each part can be assessed by selectively eliminating the stress or the neutron flux, but it is not physically possible to separate each term as discussed in [10].

Irradiation deformation models that have been developed for Zr-2.5Nb pressure tubing in CANDU reactors are either fully empirical [22,23] or semi-empirical in that there are some parameters that are derived from a mechanistic base [18,24,25]. Primary creep is empirically derived to best match available data, and it is assumed that there are three components to the steady-state creep rate, as outlined in Equation (2). The creep model published by Christodoulou et al. [24,25] aims to predict the in-reactor deformation of 6 m long tubes fabricated by extrusion and cold-working. The deformation of these tubes is important for the operation of the CANDU reactor (Figure 1a) and is complicated by the fact that the microstructure from fabrication varies along the length of the tubes such that the basal pole texture tends to be more radial and the grains tend to be thinner at the end that left the extrusion press last, the so-called back-end [10,18]. This variation in microstructure coupled with a variation in stress (σ), temperature (T) and fast neutron flux (φ) along the length of the tube complicates creep predictions. The operating temperature and fast neutron flux varies from the coolant inlet to the outlet and is different for each fuel channel. A depiction for a channel with a symmetric flux profile is shown in Figure 1b. The stress in the tubes is a function of the coolant pressure, which drops slightly from the inlet to the outlet. The hoop stress varies between about 130 MPa and 120 MPa. For a given set of operating conditions (T, φ, σ), the rate of diametral expansion is higher at the outlet because the temperature is higher. The diametral creep is also higher for the back end, so the highest diametral creep is observed when the tube orientation is such that the back end is installed at the outlet [18]. The higher diametral creep at the back end has been attributed mainly to the thinner (smaller) grains at that axial location [18,24,25]. The slip based mechanism for creep employed by Christodoulou et al. cannot account for the higher diametral creep attributed to smaller grains, and so this microstructural effect is included empirically [24,25]. The empirical model developed by Jyrkhama et al. [22] is based entirely on operating conditions for the most common pressure tube orientation (back end at the coolant outlet), although a similar model could easily be developed for the back end at the coolant inlet, which is the orientation for some reactors. Because the model from Jyrkhama et al. does not consider the pressure tube orientation, any microstructural features (texture and grain size) that are monotonically varying along the length of the tube from inlet to outlet are subsumed into the temperature coefficient.

In the model published by Christodoulou et al. [24,25], the thermal creep term is given by,
(3)$${\dot{\varepsilon }}_{d}^{\mathrm{thermal}\mathrm{creep}}=[K_{1}{C_{1}}^{d}\sigma_{1}K_{2}{C_{2}}^{d}{\sigma_{2}}^{2}]\exp(-Q_{1}/T)+K_{3}{C_{3}}^{d}\sigma_{1}\exp(-Q_{3}/T)$$

The parameters K and Q are based on best fits to available data. The anisotropy factors (C) are different for each direction, d, denoting directionality; σ_i_ are effective stresses for thermal creep. In practice, the creep response is very nearly isotropic [26]. The thermal creep shown in Equation (3) is that which applies to radiation hardened material in the creep suppression zone for the pressure tube [26]. Creep suppression is observed at very low fast neutron fluxes (between 10^14^ and 10^16^ n·m^−2^.s^−1^, E > 1 MeV) at the edges of the reactor core [27,28]. In this flux range the diametral creep rate as a function of time is an order of magnitude lower than either the out-reactor creep of unirradiated material, or the in-reactor creep of the same material.

The irradiation creep term published by Christodoulou et al. [24,25] is shown in Equation (4), and is an expression similar in form to Equation (3). The creep anisotropy factor, C_1_^d^ (*x*), varies as a function of axial location relative to the back-end (*x*) in response to changes in the texture. An additional axial correction factor, K_4_(*x*), is added to account for non-texture dependent effects, primarily the monotonic decrease in grain size along the length of the extruded tube, that affects the creep along the pressure tube.
(4)$${\dot{\varepsilon }}_{d}^{\mathrm{irrad}\text{’}n\mathrm{creep}}=K_{c}K_{4}(x){C_{4}}^{d}(x)\sigma (x)\varphi [\exp(-Q_{4}/T)+K_{5}]$$

The irradiation growth component shown in Equation (5) is given by an expression similar to Equation (4), except for the absence of a stress term. The growth anisotropy factor, K_6_(*x,*φt), is a function of axial location with respect to the back-end and fluence to take into account changes in the microstructure during irradiation that are believed to affect growth but not creep.
(5)$${\dot{\varepsilon }}_{d}^{\mathrm{irrad}\text{’}n\mathrm{growth}}=K_{g}K_{6}(x,\varphi t)\varphi \exp(-Q_{6}/T)$$

This model is complicated, having 34 adjustable parameters that are derived from many different sources. The irradiation creep part of the model is largely empirical because that part is adjusted using the K_4_(*x*) parameter derived from stress relaxation measurements in the longitudinal direction at different axial locations (defined relative to the back-end of the pressure tube) from a selection of pressure tubes [24,25].

The parameter K_4_(*x*) (which should actually be K_4_^d^ (*x*)) was derived as an adjustment to enable a fit to experimental data after applying the anisotropy parameters, C_4_^d^ (*x*), obtained from a polycrystalline model [24,25]. This means that K_4_^d^ (*x*) also accounts for any variation in texture not adequately captured by C_4_^d^ (*x*) in the polycrystalline model. In effect, K_4_^d^ (*x*) renders the anisotropy factor, C_4_^d^ (*x*) redundant, and one is essentially left with an empirical model for longitudinal creep. Having derived and applied a term for K_4_^d^ (*x*), the agreement with measurement is improved slightly by including the individual anisotropy factors, C_4_^d^ (*x*), derived from texture measurements for individual tubes rather than using a front-to-back average [25]. The improvement by using tube-specific textures is small (<10%) and is consistent with the fact that the front-to-back variation in the average anisotropy factor, C_4_^d^ (*x*), is about an order of magnitude smaller than the K_4_^d^ (*x*) empirical correction that is applied to obtain a better fit to the measurements. It might be argued that K_4_^d^ (*x*) is a magnitude parameter (the same for each direction), and that once K_4_(*x*) is settled for creep data in one direction, the anisotropy term C_4_^d^ (*x*) addresses the remaining directions. However, the evidence to support this assertion is weak. Firstly, the most important one of only four data points used to validate the model for C_4_^d^ (*x*) cannot be verified. Secondly, the polycrystalline model used to derive C_4_^d^ (*x*) assumes that the crystal can be treated as a simple cube (not even cubic close-packed), and there is no published proof that such an assumption can adequately account for the complex slip systems and Schmid factors that apply to a HCP material [24,25]. The choice of the anisotropy parameters was not based on a rigorous analysis involving the determination of the Schmid factors for each real slip system [10]; rather, anisotropy factors were chosen to best match the data at that time. Thirdly, although the model described in [24,25] assumes temperature independence for the anisotropy, more recent experimental data shows that there is a temperature dependence for the anisotropy [8,10]. More details are given in the Appendix A.

Whereas an anisotropic response to stress is one of the more important factors governing the irradiation creep of Zr-alloys, austenitic stainless steels and Ni-alloys are mostly isotropic in their response. The $B_{0}$ term in Equation (1) is equivalent to Equations (3) and (4). After first establishing a swelling rate ($\dot{S}$), which is assumed to be independent of stress, the factor D in Equation (1) is dependent on how the flux of interstitials to dislocations manifests itself as a separate creep term. The factor D is therefore equivalent to Equation (3). The net interstitial flux to the network dislocations, which is the swelling, can enable either the glide or climb of those dislocations in response to the stress. In terms of climb, the partitioning of the net interstitial flux to different orientations of dislocations in response to the stress determines the creep. For glide, the creep is like that which occurs thermally, except that instead of an excess vacancy flux, the glide is enabled by climb from an excess flux of interstitial point defects, which can be many orders of magnitude larger than the thermal vacancy flux.

### 2.2. Tensor Analysis

The compliance ($B_{0}+D\dot{S})$ relates the equivalent strain rate $\dot{\bar{\varepsilon }}$ to the equivalent stress ($\bar{\sigma })$ by Equation (1) [20,29]. The equivalent stress and strain rate are defined by,
(6)$$\bar{\sigma }=\frac{1}{\sqrt{2}}\sqrt{{\left(\sigma_{1}-\sigma_{2}\right)}^{2}+{\left(\sigma_{2}-\sigma_{3}\right)}^{2}+{\left(\sigma_{3}-\sigma_{1}\right)}^{2}}$$
(7)$$\dot{\bar{\varepsilon }}=\frac{1}{\sqrt{2}}\sqrt{{\left({\dot{\varepsilon }}_{1}-{\dot{\varepsilon }}_{2}\right)}^{2}+{\left({\dot{\varepsilon }}_{2}-{\dot{\varepsilon }}_{3}\right)}^{2}+{\left({\dot{\varepsilon }}_{3}-{\dot{\varepsilon }}_{1}\right)}^{2}}$$
where $\sigma_{d}$ and $\dot{\varepsilon_{d}}$ are the principal stresses and strain rates in each of three orthogonal directions (1, 2 and 3). When the response is isotropic, the use of these expressions allows one to compare results from different types of tests with different stress states, e.g., uniaxial and biaxial. This simple relationship cannot be applied when there is any degree of anisotropic response of the material induced by the thermo-mechanical processing [30]. For an anisotropic material, whereas the calculation of the effective stress is the same, the expression for effective strain is complicated and depends on the anisotropy factors as well as the stress state [31].

To demonstrate that $\frac{\dot{\bar{\varepsilon }}}{\bar{\sigma }}=(B_{0}+D\dot{S}$), it is easiest to start with the general anisotropic case known as Hill’s formalism. For anisotropic material, Hill derived a yield criterion equation for anisotropic material that is like the Von-Mises yield criterion for isotropic materials [32]. The expressions for effective stress and strain are shown in Equation (8) for the simple case of a stress state referred to its principal axes. Hill’s yield criterion is given by
(8)$$F{\left(\sigma_{1}-\sigma_{2}\right)}^{2}+G{\left(\sigma_{2}-\sigma_{3}\right)}^{2}+H{\left(\sigma_{3}-\sigma_{1}\right)}^{2}=k^{2}$$
where *k* is a value that defines the magnitude of the elastic strain energy needed to plastically deform a given material. It defines the size of the yield surface in stress space for a given set of anisotropy factors (*F*, *G* and *H*). This equation is the representation quadric for the tensor relating strain rate to stress that is sometimes referred to as Hill’s compliance tensor, although Hill never once mentioned tensors in his book [32].

In terms of a tensor equation for the creep strain rate as a function of the stress state, the formal definition of the compliance (referred to the principal axes of stress) is,
(9)$$\left(\begin{array}{c} \dot{\varepsilon_{1}}\\ \dot{\varepsilon_{2}}\\ \dot{\varepsilon_{3}}\; \end{array}\right)=f\left(M,\Phi ,T\right)\cdot \left(\begin{array}{ccc} F+H & -F & -H\\ -F & F+G & -G\\ -H & -G & H+G \end{array}\right)\cdot \left(\begin{array}{c} \sigma_{1}\\ \sigma_{2}\\ \sigma_{3} \end{array}\right)$$
where $f\left(M,\Phi ,T\right)$ is a scaling factor (sometimes referred to as a compliance), which is also dependent on the magnitude chosen for *F*, *G* and *H*, relating the strain rate to the stress state $\left(\sigma_{1},\;\sigma_{2},\sigma_{3}\right)\;$for a given temperature (T), neutron flux (Φ) and material or microstructure (*M*). The characteristic equation giving the shape of the tensor in stress space (the yield surface) is given by,
(10)$$\sigma^{T}K\sigma =\left(\sigma_{1}\;\sigma_{2}\;\sigma_{3}\right)\cdot \left(\begin{array}{ccc} F+H & -F & -H\\ -F & F+G & -G\\ -H & -G & H+G \end{array}\right)\cdot \left(\begin{array}{c} \sigma_{1}\\ \sigma_{2}\\ \sigma_{3} \end{array}\right)$$
giving
(11)$$\sigma^{T}K\sigma =\;F{\left(\sigma_{1}-\sigma_{2}\right)}^{2}+\;G{\left(\sigma_{2}-\sigma_{3}\right)}^{2}+\;H{\left(\sigma_{3}-\sigma_{1}\right)}^{2}$$

A plot of Equation (8) in stress space gives a surface, in this case a cylinder. If *F*, *G* and *H* are chosen so that *k* = 1 in Equation (8), it is easy to see that k derived from Equation (9) becomes $\frac{1}{\sqrt{f\left(M,\Phi ,T\right)}}$, which, together with *F*, *G* and *H*, determines the size of the surface in stress space, in this case the yield surface (a cylinder). Given that *F* + *H*, *F* + *G* and *H* + *G* represent the tensor properties for each principal direction of a symmetric tensor, they must satisfy Equations (8) and (9). The form of the tensor in Equations (9) and (10) is also the condition for constant volume, i.e., the sum of the strain rates in the principal directions are zero for any stress. The magnitude of *F*, *G* and *H* can be chosen for convenience because any scaling of *F* + *G* + *H* is subsumed into the compliance term, $f\left(M,\Phi ,T\right)$. It has been stated that *F* + *G* + *H* = 1.5 [25]. This could possibly lead one to believe (mistakenly) that if one knows two of the parameters then the other is defined by that arbitrary summation. To do so would be incorrect. It is clear from the above that the choice of scaling factor for *F* + *G* + *H* should be used with the understanding that F, G and H still must satisfy Equations (8) and (9).

In terms of a tensor equation for creep strain rate, one can define the compliance for an isotropic material (referred to the principal axes of stress) by making the anisotropy parameters equal, $F=G=H=\frac{1}{3}$, so that *F* + *G* + *H* = 1, for example, and then equating the strain rate to stress,
(12)$$\left(\begin{array}{c} \dot{\varepsilon_{1}}\\ \dot{\varepsilon_{2}}\\ \dot{\varepsilon_{3}}\; \end{array}\right)=\left(B_{0}+D\dot{S}\right)\cdot \left(\begin{array}{ccc} \frac{2}{3} & -\frac{1}{3} & -\frac{1}{3}\\ -\frac{1}{3} & \frac{2}{3} & -\frac{1}{3}\\ -\frac{1}{3} & -\frac{1}{3} & \frac{2}{3} \end{array}\right)\cdot \left(\begin{array}{c} \sigma_{1}\\ \sigma_{2}\\ \sigma_{3} \end{array}\right)$$
where $\left(B_{0}+D\dot{S}\right)$ is the magnitude of the creep compliance relating the magnitude of the strain rate to the stress state $\left(\sigma_{1},\;\sigma_{2},\;\sigma_{3}\right)$, referred to the principal axes. As the swelling rate ($\dot{S})$ will vary with neutron flux, fluence and temperature (assuming it is independent of stress), the creep compliance $\left(B_{0}+D\dot{S}\right)$ is expected to vary as the swelling evolves because $\dot{S}$ is not constant. It turns out that neither is $B_{0}$, both being governed by the microstructure evolution. At any one point in time, however, for any stress state it is easy to show that,
(13)$$\dot{\varepsilon_{1}}=\frac{\left(B_{0}+D\dot{S}\right)}{3}\left(2\sigma_{1}-\sigma_{2}-\sigma_{3}\right)$$
(14)$$\dot{\varepsilon_{2}}=\frac{\left(B_{0}+D\dot{S}\right)}{3}\left(-\sigma_{1}+2\sigma_{2}-\sigma_{3}\right)$$
(15)$$\dot{\varepsilon_{3}}=\frac{\left(B_{0}+D\dot{S}\right)}{3}\left(-\sigma_{1}-\sigma_{2}+2\sigma_{3}\right)$$

From which it follows that
(16)$$\left({\dot{\varepsilon }}_{1}-{\dot{\varepsilon }}_{2}\right)=\left(B_{0}+D\dot{S}\right)\cdot \left(\sigma_{1}-\sigma_{2}\right)$$
(17)$$\left({\dot{\varepsilon }}_{2}-{\dot{\varepsilon }}_{3}\right)=\left(B_{0}+D\dot{S}\right)\cdot \left(\sigma_{2}-\sigma_{3}\right)$$
(18)$$\left({\dot{\varepsilon }}_{3}-{\dot{\varepsilon }}_{1}\right)=\left(B_{0}+D\dot{S}\right)\cdot \left(\sigma_{3}-\sigma_{1}\right)$$
and thus,
(19)$$\frac{\frac{1}{\sqrt{2}}\sqrt{{\left({\dot{\varepsilon }}_{1}-{\dot{\varepsilon }}_{2}\right)}^{2}+{\left({\dot{\varepsilon }}_{2}-{\dot{\varepsilon }}_{3}\right)}^{2}+{\left({\dot{\varepsilon }}_{3}-{\dot{\varepsilon }}_{1}\right)}^{2}}}{\frac{1}{\sqrt{2}}\sqrt{{\left(\sigma_{1}-\sigma_{3}\right)}^{2}+{\left(\sigma_{2}-\sigma_{3}\right)}^{2}+{\left(\sigma_{3}-\sigma_{1}\right)}^{2}}}=\frac{\dot{\bar{\varepsilon }}}{\bar{\sigma }}=\left(B_{0}+D\dot{S}\right)$$

This shows why the creep compliance for an isotropic material is only dependent on the ratio of the effective strain rate and effective stress. For the anisotropic case, one must determine *F*, *G* and *H* relative to the component coordinate system by testing and measuring the creep strain rate of tensile specimens along each of the three principal axes, which gives *F* + *H*, *F* + *G* and *H* + *F*. This is often not possible to do for a component such as a thin-walled tube such as a flux thimble or fuel pin. One can obtain enough information to determine *F*, *G* and *H* from two tensile specimens (perhaps even bent beam stress relaxation specimens) if one can simultaneously measure the strain rate in both the long and short axes of at least one specimen [10]. Performing tests to obtain *F*, *G* and *H* relevant to the component of interest is difficult in an irradiation environment.

Given that $\frac{\dot{\bar{\varepsilon }}}{\bar{\sigma }}=(B_{0}+D\dot{S}$), the study of the irradiation creep of isotropic material focusses on the separate coefficients of the compliance, $B_{0}$ and $D\dot{S}$. The $B_{0}$ term is most important prior to the onset of swelling.

The effect of the microstructure on irradiation creep will now be assessed separately in terms of dislocation loops, network dislocations, and grain structure.

## 3. Microstructure Effects on Irradiation Creep

### 3.1. Effect of Prismatic Dislocation Loop Evolution on Irradiation Creep

Dislocation loop evolution is most important in the early stages of irradiation because of the rapid increase in loop density that occurs from point defect clustering. Loop formation occurs in both annealed and cold-worked materials, although loop nucleation and evolution can be aided in cold-worked material from the helical climb of screw dislocations [10].

For creep at high temperatures, recovery of the dislocation network structure is significant, and the lower (thermal) creep rates that have been reported at high temperatures are the result of lower dislocation densities due to thermal recovery in stainless steels [5,33,34]. At lower temperatures, when recovery of any cold-worked dislocation structure is not apparent, there is often suppression of dislocation creep because of radiation damage clustering [8,10], mainly from dislocation loop formation.

It seems that some form of stress-induced climb and glide (SICG) mechanism [8,10,35] is probably dominant when it comes to explaining the swelling-independent B_0_ term. The effect of hardening in reducing irradiation creep by glide is illustrated from an analysis of the 316 SS data of Lewthwaite and Mosedale [36]. Creep measurements from spring relaxation at low doses were obtained for various stresses and for a range of neutron fluxes over a narrow temperature range. By taking the slope of the curve from a plot of strain as a function of dose and dividing it by the stress, one obtains the creep compliance (*B*_0_) at each stage of the creep evolution provided it is assumed (reasonably) that the creep strain would be zero at zero stress. This is only true provided the material is not swelling and exhibits no irradiation growth [37]. Taking the Lewthwaite and Mosedale data [36] and plotting the compliance against dose shows that the secondary (“steady-state”) creep compliance (*B*_0_) decreases as the dose increases and reaches a true steady state (constant) condition after about 5–10 dpa for the various austenitic stainless steels [8]. For 316 stainless steel, the steady-state compliance is about 3 × 10^−6^ MPa^−1^.dpa^−1^ at doses > 5 dpa, Figure 2. The saturation in the compliance at a given dose can be associated with the saturation in yield stress at a dose of about 5–10 dpa that has been measured for 316 stainless steels irradiated in the BOR 60 reactor at 330 °C, see inset [38]. For most materials, B_0_ varies from 0.5 to 4 × 10^−6^ MPa^−1^.dpa^−1^ [8,39].

According to Garner [5], part of the difficulty in measuring *B*_0_ in the swelling regime is in separating the swelling strain and the irradiation creep strain at high doses. For the Lewthwaite and Mosedale data [36], the irradiation dose is low enough that swelling can be ignored and the measured creep can be attributed to B_0_ only. The factor *B*_0_ can be considered as that part of the steady-state irradiation creep that comes from dislocation slip via the stress-induced climb and glide (SICG) mechanism or simply climb, both of which may be enhanced by the effect of stress on diffusional mass transport [8].

By citing results presented by Lewthwaite and Proctor [40], where they showed a non-linear relationship of D/D_e_ with dose (D is the creep deflection and D_e_ the initial elastic deflection) of a weighted spring, Garner and Tolocyko [41] unfairly claimed that the results of Lewthwaite and Mosedale [36] on secondary steady state creep in the absence of swelling (*B*_0_) had been misinterpreted because they had not considered the transient, which was an artefact of the data analysis by Lewthwaite and Proctor. However, Lewthwaite and Mosedale [36] measured creep rates starting about 1000–2000 h after the start of the irradiation, and did not include the primary transient, which is a combination of both the elastic and anelastic deflection of the weighted spring and any additional transient that is observed during in-reactor creep tests up to about 2000 h [26]. For each set of measurements [36], the creep was approximately linear as a function of dose (at a constant dose rate) up to 30,000 h (about 30 dpa). If there is a criticism to be made of Lewthwaite and Mosedale it is in the approximation that the secondary creep with dose was linear after an initial transient, when in fact it only becomes linear after the evolving dislocation loops have saturated (after 5–10 dpa). Garner and Tolocyko [41] confounded the two sets of results, the transient in the Lewthwaite and Proctor [40] data being an artefact of dividing D (the deflection) by D_e_ (the initial offset), the latter being unrelated to the creep. The measurements of Mosedale and Lewthwaite [36] apply to the post-offset creep only and were not confounded by any initial offset giving rise to the transient in D/D_e_ shown by Lewthwaite and Proctor [40]. The Mosedale and Lewthwaite [36] data show that the slope of the creep versus dose curve (*B*_0_) decreases with dose and approaches a constant value after a certain low dose that can be attributed to the hardening of the material due to the evolution of the dislocation loop structure [8,10].

The saturation in *B*_0_ at 5–10 dpa (Figure 2) is consistent with the evolution of the radiation damage dislocation loop structure, as evidenced by yield strength measurements in 316 SS (see inset [38]). Although some degree of climb will inevitably contribute to the steady-state creep behaviour at doses > 5–10 dpa, the reduction rather than augmentation of the irradiation creep rate with increasing dose implies that dislocation glide is the dominant creep mechanism up to about 5 dpa. Given that the creep is likely determined by the climb and/or glide of dislocations, how much of the ensuing creep at doses > 5 dpa can be attributed to either strain mechanism cannot be discerned without more data. Ultimately, in terms of magnitude, climb can be considered equally important as glide as a strain determining mechanism. One has only to consider what swelling is, i.e., the manifestation of an excess of interstitials that migrate to sinks other than cavities (in this case dislocations) to realise that strain rates from interstitial diffusional alone are not necessarily small. The requirement for a direct creep dependence (the deviatoric strain) on diffusional mass transport is that stress has to have an effect on the anisotropy of diffusion [42], or that stress affects which dislocations climb more than others, i.e., stress-induced preferential absorption (SIPA) [43]. Either way, the evolution of the B_0_ compliance with increasing dose can be reasonably attributed to the evolution of the dislocation loop structure that, in effect, suppresses creep from dislocation slip and reduces the stain rate to a level that may then be dictated by diffusional mass transport governed by the atomic displacement damage rate, which is a function of the neutron flux and spectrum [44,45]. The best way to see what this means during irradiation is to compare thermal (out-reactor) creep rates of material that is both pre-irradiated and unirradiated.

Swelling-independent creep also applies to Zr-alloys that have negligible swelling, even at high doses. Zr-alloys are different in that they are anisotropic, but there are many commonalities with austenitic alloys that are in the non-swelling regime. The main difference between Zr-alloys and other austenitic engineering alloys is that they exhibit irradiation growth because of the intrinsic crystallographic anisotropy, Zr-alloys being hexagonal close-packed (HCP) compared with face-centred-cubic (FCC) austenitic alloys. Irradiation growth is not unique to HCP metals, as it can also be exhibited when the microstructure (dislocations in particular) is anisotropic because of fabrication [37]. The intrinsic radiation response (irradiation growth) of Zr alloys itself evolves as the microstructure evolves. In this case, the appearance of vacancy basal plane dislocation loops in the microstructure (either discrete loops or from climb on c+a screw dislocations) leads to shrinkage along the c-axis, irrespective of the stress state [10,37], and is manifested as a non-linear strain when irradiation growth is plotted against the irradiation dose [10,37]. The growth itself is manifested as a non-zero intercept at zero stress in plots of irradiation creep rate against stress [10].

Figure 3 shows diametral strains as a function of time derived from pressurised creep capsule data reported by DeAbreu et al. [26] for identical (within a small range of manufacturing variability) creep capsules. The diametral strain of the pressurised capsules was measured both in and out of reactor at low and high fluxes, of the order of 10^16^ n·m^−2^·s^−1^ and 10^17^ n·m^−2^·s^−1^ (E > 1 MeV), respectively. For zirconium alloys the displacement damage rate is proportional to fast flux and, in the NRU spectrum, 10^17^ n·m^−2^·s^−1^ is equivalent to 1.635 × 10^−8^ dpa·s^−1^, where dpa refers to the primary damage [45].

In terms of freely migrating defects (FMDs), the creep rate is further reduced by spontaneous recombination within collision cascades and, as we shall see later, the calculated FMD ratio for Zr-alloys that best fits creep data is about 0.07. This number is slightly larger than those (ranging from 0.01–0.03) that have been deduced from rate theory calculations to match growth data [10], but the scaling one needs to apply for the FMD fraction depends on the choice of the other parameters used in the model. Similar capsules that were either unirradiated or had been previously irradiated to high doses were tested at the same temperature in the laboratory for comparison. It is apparent from Figure 3 that irradiation at low dose rates suppresses creep compared to an out-reactor creep test of the same material. The suppression of the creep rate (as a function of time) is more pronounced as the dose increases and more dislocation loops are generated, until a saturation in loop density is achieved and pseudo-steady-state conditions then prevail [8,9]. The other interesting aspect of the data plotted in Figure 3 is that there is a complete suppression of any primary creep when testing pre-irradiated material. The primary creep is apparent for the unirradiated material, with the steady state thermal creep rate approaching that of the pre-irradiated material at long times. The hardening in that case can be attributed to conventional work-hardening. This behaviour is consistent with the primary creep being a manifestation of dislocation slip, which is suppressed in the pre-irradiated material.

From Figure 3 it appears as if the primary creep is over after about 2000 h. This is deceptive because the creep rate is constantly changing; it is just very rapid at the beginning of the irradiation because the dislocation loop structure is evolving rapidly and because the helical climb on screw segments of network dislocations renders those dislocations immobile [7,8,9,10]. For Zr-alloys there are two types of loops: (i) a-type that are small and saturate at low doses (<4 × 10^24^, E > 1 MeV or <0.6 dpa); and (ii) basal c-component loops that are large and evolve over a longer time/dose range. The two types of radiation damage in Zr-2.5Nb pressure tubing are shown in Figure 4.

The TEM micrographs are difficult to interpret quantitatively, and thus X-ray diffraction is used to monitor the evolution of the two types of radiation damage. The line broadening of prism planes (a-type loops) or basal planes (c-component loops) has been used for many years to account for the physical property changes of irradiated pressure materials [46,47]. One can use the integral breadth line broadening measurements to show dislocation loops evolution assuming that the increase in integral breadth is directly correlated with either a-type loops (prism plane broadening) or c-component loops (basal plane broadening).

With respect to irradiation creep, the a-type loops are like unfaulted Frank loops that cause irradiation hardening in austenitic stainless steels [5,6]. The faulted, sessile c-component loops that evolve in cold-worked Zr-2.5Nb pressure tubing are important because they nucleate and grow by helical climb on c+a dislocations having a screw character. The helical climb itself contributes to the strain but, perhaps more importantly, these loops are sessile and thus render the screw components of the c+a dislocation network immobile. There is little evidence that the c-component dislocation network, which contains many sessile junctions, changes with irradiation other than by helical climb [48], so it is unlikely that the dislocation glide of c+a dislocations contributes to irradiation creep in these cases. The climb mechanism can be illustrated using in situ electron irradiation. Figure 5 shows the helical climb of c+a screw dislocations in Zr during electron irradiation.

The evolution in the creep response at high doses is apparent from published creep rates for creep capsules [26,49] and pressure tubing [50], and is plotted in Figure 6 [10]. It is clear from Figure 6 that the irradiation creep rate as a function of dose is continually decreasing over a large dose range. The initial decrease that occurs over a dose range that is less than 1 dpa (<6 × 10^24^ n·m^−2^, E > 1 MeV) can be attributed to the evolution of the a-type dislocation loop structure (see inset). The longer duration of decreasing creep rate that is observed at high doses (circled area) can be attributed to the evolution of the c-component dislocation structure (helical climb) shown in Figure 4, and is consistent with rate theory modelling results [10].

After an initial rapid increase in line broadening resulting from a-type loop formation, the slower upward trend in prism plane line broadening (shown in the inset) is matched by the slow increase in basal plane line broadening, and both features can be attributed to the c-component dislocation loop evolution, the loops having an a- and c-component in their Burgers’ vector ($\frac{c}{2}+p$). The decreasing creep rate at high doses can thus be attributed to the long-term evolution of the basal c-component loop structure that both inhibits glide (if one assumed that dislocation slip is a contributor to the strain) but also adds a negative component to the diametral strain. The negative strain associated with the helical climb is because the basal plane loops are consistently vacant in nature, and because the tubes have a strong transverse basal texture, so that there is a large component of the c-axis (shrinkage) strain directly from the loops resolved in the transverse direction of the tube [10].

There is a strong temperature dependence for the loop formation, as evidenced from the line broadening. Figure 7 shows prism plane line broadening data for Zr-2.5Nb irradiated in the OSIRIS reactor at 250 °C [51], and also data for a Zr-2.5Nb pressure tube from assembly 50206 irradiated in the NRU reactor at about 275–300 °C [28]. There is less variability in the OSIRIS data because they come from standard fracture toughness specimens that were taken from the same location of one pressure tube and are thus subject to less spatial, and tube-to-tube variability, that normally exists. A temperature range is given for the NRU case because that part of the tube where measurements were made was irradiated for the first 52 kh at 275 °C, and then from 52 kh to 67 kh at 300 °C. This temperature shift during operation was because the coolant flow in the reactor was modified for that channel later in life. The important point to note is that the lower irradiation temperature in OSIRIS (250 °C) corresponds with higher values for XRD line broadening, which is interpreted as a higher a-type dislocation loop density. There is no discernible effect of damage rate because the line broadening from ex-service CANDU pressure tubing overlaps with the OSIRIS data, obtained with a damage rate that is about 10 times greater than for the CANDU reactor at the same irradiation temperature [47,51]. The points for the CANDU reactor data taken from measurements of more than 20 tubes after irradiation at temperatures between 250 °C and 260 °C are also shown in Figure 7. There is more scatter compared to the other data because the test reactor data (NRU and OSIRIS) apply to individual tubes sampled over a small axial distance with minimal variation in the as-fabricated microstructure. The effect of temperature on dislocation loop density is consistent with the lowering of the steady-state irradiation creep rate at lower temperatures because of higher loop densities (see Section 3.4).

For CANDU reactors in the main body of the reactor core, the operating temperature ranges between about 250–300 °C. The effect of temperature on creep rate can be demonstrated by comparing available creep data from two NRU fuel assemblies at 275 °C and 300 °C (Figure 8). The data are shown together to illustrate that the pseudo steady state creep rate at 275 °C (0.1% dpa^−1^) is lower than that at 300 °C (0.2% dpa^−1^). The creep rates have been converted to % per dpa for ease of comparison with data from 316 stainless steel, as shown in Figure 2. For the austenitic stainless steel irradiated at temperatures between 247 °C and 360 °C, the steady state shear creep compliance is approximately 4 × 10^−6^ dpa^−1^·MPa^−1^. Taking the steady-state strain rates shown in Figure 8 and dividing them by a hoop stress of about 120 MPa (assuming growth is negligible), the compliance would be between about 1 and 2 × 10^−5^ dpa^−1^·MPa^−1^ for the CANDU pressure tubing at 275 °C and 300 °C, respectively, which is larger than for the shear creep compliance of the stainless steel.

Part of the difference in the steady-state creep rate when comparing the cold-worked 316 stainless steel spring data with the cold-worked Zr-2.5Nb pressure tube data may be attributed to the different stress states—a shear compliance for the DFR springs and a biaxial tensile compliance for the Zr-2.5Nb pressure tubing. The larger compliance for the CANDU pressure tubing may also be a result of a more complex microstructure, small platelet grains that are <0.6 μm thick, and a high dislocation density. No further comparison is possible without a detailed assessment of the stress states and microstructures in each case. The important point is that both stainless steel and Zr-2.5Nb pressure tubes exhibit the same trend in creep with increasing dose at doses < 10 dpa. They each show a decrease in irradiation creep rate that drops to a low steady-state value that corresponds with the evolution and then saturation of the dislocation loop structure occurring over a dose range up to about 0.5 dpa for Zr-2.5Nb and 5–10 dpa for the 316 stainless steel. The larger dose needed before steady-state (secondary) creep is achieved for the 316 stainless steel can be attributed to the fact that steels have a much larger vacancy migration energy than Zr-alloys, and thus post-cascade recombination is likely larger, further reducing the number of FMDs available for microstructure evolution and slowing the rate of dislocation loop evolution [7,9].

The effect of dislocation loop evolution is most easily seen at low doses, affecting the low dose creep rate. At high doses, only for c-component loops in Zr-alloys is there an observable effect of the c-component density, and this is because those defects evolve over a large dose range. Once the Frank loops in austenitic alloys and a-type loops in Zr-alloys have reached a pseudo steady-state condition, their effect on creep (as barriers to slip and sinks for point defects) is expected to be more-or-less constant.

One of the issues with slip-based creep models in small-grained material such as Zr-2.5Nb pressure tubing is the fact that the network dislocations should eventually be exhausted and the creep rate should diminish after strains that should be <1% in the case of small-grained material such as Zr-2.5Nb pressure tubing [10]. This is an important point that would help distinguish between the different contributions to creep because one would expect creep from dislocation slip to be smaller with decreasing grain size. In fact, the opposite is true [10,18], bringing into question the assumption of dislocation slip in some creep models [24,25]. A slip-based mechanism could be invoked to help explain the observed decreasing rate at high doses shown in Figure 6 were it not for the observations of c-component loop evolution that can account for the lower strain rate simply from loop growth.

There is no evidence for lower creep rates at higher doses that would be consistent with exhausting slip dislocations, and neither is there any evidence to show that irradiation creep is necessarily larger for cold-worked compared to annealed material [8]. To address this enigma, if one assumes that all irradiation creep has to be determined by dislocation slip, it has been suggested [5,6,8] that dislocation loops can evolve into a glissile dislocation network that would otherwise be absent for annealed material or depleted by the creep of cold-worked material at high doses. Although loop evolution to the extent envisaged has only been observed at very high doses and high irradiation temperatures in stainless steels [5,6], the conjecture is that loop evolution can provide a mechanism for creep by the dislocation slip of annealed material and the continuous replenishment of a glissile network for cold-worked material. The argument posed fails to account for the fundamental difference between the climb and glide of prismatic dislocation loops and network dislocations, the latter being created by yielding. In the prismatic loop case, the strain is realised as soon as it is produced. For a network dislocation (produced by yielding), the strain is a function of the distance the dislocation has travelled from the point at which it was created, and is given by ρbl, where ρ is the dislocation density (m.m^−3^), b is the Burgers vector of the dislocations (m), and l is the distance travelled (m). The main flaw in the regeneration argument proposed in [8] stems from the way in which the distortions of the planes by dislocations have been perceived.

In the 2020 creep review by Onimus et al., [8], a creep mechanism was described whereby prismatic dislocation loops grow and form a network that can glide and create a plastic shear strain that is in addition to the dilatational strain from the loop itself. Whereas the shearing of a prismatic loop within its glide cylinder that redistributes the loop strain is conceivable, the mechanism proposed in [8] results in the creation of additional strain that, in retrospect, is unphysical. There were two problems with what was proposed (see Appendix A) [8]: (i) the glide direction did not conform to the work done by the applied shear stress; and (ii) the elastic strain distortion observed at the boundary surface of the single crystal was not allowed to relax as each dislocation half plane moved away from the surface. This prevention of relaxation of the distorted atomic planes as the dislocations moved resulted in an unphysical lack of response for the distorted planes given that the reason for their distortion had been removed. There may be a perception that the shear stress somehow holds the depicted distortion of the lattice planes in place, but this is also misperceived. In the first instance, creep stresses are very small and the elastic strain << the plastic strain that exists near the core of a dislocation. The applied stress in the creep process gives a gentle push in one direction that can eventually allow the dislocation to move given that it is thermally vibrating [1]. If one considers a shear stress of 50 MPa (say), and a shear modulus of 35 GPa, the elastic strain is <$\frac{1}{10}$th of a degree, an imperceptible shear distortion relative to the large distortion depicted in Figure 10 of [8].

In addition, apart from the unphysical nature of the proposed mechanism, any dislocation movement in a polycrystalline specimen would be against the constraints from neighbouring grains. Even though the material in neighbouring grains will be elastically distorted by similar small amounts, the plastic step envisaged at the edge of the grain would be equivalent to the creation of a Stroh-type crack [52], requiring an applied stress that is of the order of GPa, not the small stresses that apply during creep.

For Zr-alloys, the loop evolution that does occur at high doses involves sessile basal plane c-component loops. In annealed material, the strain from the c-component loops is not apparent until the loops have nucleated and then grown to detectable sizes, giving the appearance of an incubation period before the effect is manifested as a measurable strain. For cold-worked Zr-alloys, there is no such period because the c+a screw dislocations act as easy nucleation sites for the c-component loops from the very beginning of the irradiation [53]. As these loops are faulted, they are both barriers to slip of the existing network and prevent any glide of the c+a screw dislocations. It may be possible for edge c+a dislocations to glide, but it is unlikely given that in cold-worked material the c-component dislocations typically form a network with many of the junctions being effectively sessile [54].

### 3.2. Effect of Network Dislocation Density on Irradiation Creep

Data showing the dependence of irradiation creep on dislocation density (cold-working) is ambiguous. Although the dislocations are important as sinks for point defects, their role in promoting creep through SICG is not obvious. An examination of the data in reference [8] shows that there is little difference in the value of the swelling-independent creep compliance ($B_{0}$) between cold-worked and solution-annealed austenitic stainless steels in the temperature range of 300–400 °C, which is in the sink-dominated regime for most engineering materials where one expects irradiation effects to dominate material behaviour. Similar insensitivity to cold-working has been reported by Garner [6]. This is not surprising if one assumes that the creep rate is a function of diffusional mass transport because point defect fluxes to dislocation sinks are independent of temperature between 300–400 °C for a vacancy migration energy of about 1.4 eV [7,8,9]. Grossbeck et al. [55] showed that, for specific austenitic alloys, the irradiation creep of cold-worked material is indeed higher than for annealed material at low temperatures (60 °C), Figure 9. Coupled with the fact that, for the cold-worked material, the irradiation creep at 60 °C is higher than creep at 330–400 °C, one can conclude that dislocation glide is easier at lower temperatures in the range from 60–400 °C for austenitic alloys. One explanation for this temperature dependence is that the point defect fluxes are so low that loop clustering does not occur, and the low-temperature creep is dominated by dislocation slip unhindered by dislocation loops. This is consistent with the observation that there is little evidence of radiation damage clustering (other than He bubble formation) in Inconel X-750 irradiated in a CANDU reactor at temperatures < 200 °C [7]. This inverse temperature dependence has also been observed for the stress relaxation of Inconel X-750 and CW 304 stainless steel, with in-reactor creep rates being faster at 60 °C compared with 300 °C [7].

It may be expected that the cold-working of a given alloy should increase the thermal creep at high enough stresses and temperatures [4]. The same could also be expected for irradiation creep (if the dominant deformation mechanism is SICG).

One of the peculiarities of the effect of cold-work on irradiation creep at moderate stresses and temperatures is that it does not scale with the amount of cold work/dislocation density. This has been demonstrated for Zr-2.5Nb [49] and also for 316 austenitic stainless steel [56]. The results of Garner et al. [56] showing diametral creep strains of pressurised creep capsules as a function of dpa for different stresses (presumably hoop stresses) have been reproduced in Figure 10a. The strain rates for doses >25 dpa plotted as a function of cold-work are shown in Figure 10b. The strain rates were obtained from a linear fit to the data because the 10% and 20% cold-worked cases indicated that a linear fit was the best choice. Even though the degree of cold-working has increased three-fold, the creep rates are indistinguishable at high doses. A linear extrapolation of the strain as a function of dose in Figure 10a indicates a possible negative offset of the dimensions, and has been attributed to densification [5,6], but could also be the result of a systematic measurement error (of a standard for example). This type of offset has been observed in other creep tests by Garnier et al. [57]. Quite apart from showing a similar offset to that shown in Figure 10a, the results of Garnier et al. [57] show that the compliance (slope with stress) is also qualitatively similar regarding cold-working, i.e., the absolute strain is larger (less negative offset) for solution annealed material compared with cold-worked material. The strain rate results of Garnier et al. [57] also show a similar non-linear trend with stress to that shown in Figure 10b. Other than densification cited by Garner [5,6], the reason for the dose offset and the non-linearity with stress is not known. It is unlikely to be associated with the incubation that is often observed before the onset of swelling [5,6] because, for the data in Figure 10, the swelling is negligible up to about 40 dpa and very small even at 60 dpa (see 0 MPa data in Figure 10). It is possible that dislocation slip is more prominent at high stresses (200 MPa), and that could, in part, account for the curvature shown in Figure 10b.

More recent data from pressurised creep capsules irradiated in the NRU reactor are shown in Figure 11. These data were generated as part of the Atomic Energy of Canada Limited (AECL) nuclear platform program to support fundamental research and were presented at the 18th International Symposium of Zr in the nuclear industry and published in the proceedings by DeAbreu et al. [26]. Plots of the steady state creep rates against stress show the effect of stress, irradiation temperature and the effect of changing the cold-work from 12% to 27%. It is clear from these data that the compliance (slope of the strain rate against stress) is not constant at a high temperature (340 °C), indicating that the stress exponent is >1 at that temperature. The slope appears constant at lower temperatures, indicating a stress exponent = 1. Increasing the cold-working has a marked effect on the creep rate at 340 °C, which together with a non-linear stress exponent indicates that dislocation slip is the dominant mechanism at that temperature. Although the 27% cold-worked material has a higher creep rate at 280 °C and 320 °C, the slopes of the curves are linear, suggesting that n = 1, which is generally accepted as the condition for diffusion-controlled creep [4,58].

The lack of sensitivity of the irradiation creep compliance (slope of the creep rate versus stress curve) to the dislocation density shown in Figure 11 for cold-worked Zr-2.5Nb at temperatures of 280 °C and stresses < 200 MPa is consistent with results from OSIRIS [49]. This insensitivity of irradiation creep to dislocation density at stresses < 200 MPa is also manifest for Zircaloy-2 irradiated at low temperatures (about 60 °C) in NRU [59]. Whereas there is a difference in absolute creep rates for the 12% and 27% cold-worked materials, it can also come about from differences in the amount of dislocation climb from simple mass transport. These results, coupled with those of Garner et al. [56] shown in Figure 10, cannot be easily reconciled with the notion that irradiation creep is dominated by dislocation slip and simply enabled by the elevated point defect concentrations during irradiation. Neither are they consistent with a pure mass transport mechanism for strain. Rather, they indicate that some mixing of the two mechanisms is likely, with slip dominating at the higher temperatures and stresses. The corollary to this is that irradiation creep is dominant at low temperatures and stresses. For secondary (steady-state) creep of Zr-alloys, diffusional mass transport appears to be dominant at stresses < 150 MPa and temperatures < 300 °C. For austenitic stainless steels, diffusional mass transport appears to be dominant at 400 °C and stresses below 200 MPa (see Figure 10).

The fact that there is a somewhat contradictory insensitivity to the amount of cold-work in the “steady-state”, as discussed by Onimus et al. [8], leads one to hypothesise that there are two factors at play concerning the role of network dislocations in being sources for strain through: (i) glide and (ii) climb. Irradiation can both enhance creep by creating new sinks for point defects in the form of dislocation loops at low doses and cavities at high doses; there is also a suppressing effect of these defect clusters by inhibiting glide. From the perspective of mass transport, it is the ratio of the sinks rather than their absolute magnitude that is important [7,8,9,10,12]. In this context, a change in the ratio of the sink strengths for different point defects is important. It is worthwhile pointing out that, if the irradiation creep was dominated by dislocations and mass transport, it is not the magnitude of the sinks that is important, rather it is the ratio of the biased sinks. In this case, for a fixed number of available interstitial point defects, if there was a stress-induced bias favouring interstitial diffusion to dislocations of one orientation compared to another, the creep rate (after having subtracted swelling) would be dependent on that bias rather than on how many dislocations there were. In the sink-dominated regime, the irradiation creep is also controlled by the rate at which the point defects are created, rather than the speed with which they arrive at the sinks. The irradiation creep that is in question is a constant volume process, and a positive strain in one direction (due to the stress), whether by slip or mass transport, must be matched by a negative strain in another direction. At any instant in time, one therefore has a certain number of interstitials that are dependent on the ratio of the cavity/void density and the dislocation density that migrate to dislocations defining the swelling. In the absence of a stress, these interstitials have no preference for which dislocation orientation they migrate to, and isotropic swelling results. If the interstitial flow to dislocations was itself biased by the action of the stress, one would observe some anisotropy, which is the creep. Irradiation creep, which is a zero-volume process, can be regarded as a distortion of the swelling, i.e., there is only so much excess matter available to be manifested as strain by the absorption of interstitials at dislocations resulting in strain from climb alone or from climb and glide. If the creep is controlled by the flux of interstitials to dislocations that are affected by a stress bias, then the creep, i.e., that fraction of the swelling that is distorted by the stress, may be expected to be independent of the dislocation density (see Section 3.3). If the creep is simply the activation of dislocation motion in response to a stress, albeit aided by the point defect flux, one might expect to see a dependence of the creep rate on the dislocation density.

### 3.3. Effect of Cavity Swelling on Irradiation Creep

Given that the non-swelling component of the irradiation creep compliance of austenitic alloys (*B*_0_) is governed by the same considerations that apply to Zr-alloys, i.e., dislocation slip, and how it is shut down by the formation of dislocation clusters, the other component of irradiation creep that is pertinent to austenitic alloys is that governed by swelling. In the creep equation used to describe the irradiation creep of austenitic alloys (Equation (1)), the compliance D is, by definition, a factor used to link the irradiation creep rate to the swelling rate. It is experimentally difficult to ascertain how much of the dependence on the swelling rate is simply because of the linear strain from swelling and how much is because of a true enhancement of the creep, i.e., whether D is effectively a multiplier of whatever the linear swelling would have been in the absence of a stress. This would be true if the creep represented by D was the result of dislocation slip because the more interstitial point defects are available to assist the climb and glide process the higher the creep rate would be from the dislocation slip. An alternate view is that, if the creep was simply the distortion of the swelling (swelling being the deposition of interstitial point defects at different sinks), then D amounts to an anisotropy parameter that partitions the available interstitial point defects amongst sinks of different orientations, the partitioning and thus the anisotropy being dictated by the stress state. Under those circumstances, the creep may be expected to be independent of the number of dislocations. However, the swelling is not independent of the dislocation density, so the creep compliance (D) and swelling rate (S) are not independent. Anything that increases the dislocation bias (in this case a stress-induced bias) may lead to an increase in the swelling rate and conceivably an augmentation of the swelling (and thus creep) simply from the stress itself. The question arises how does one interpret Equation (1) when the swelling and the creep rate are inter-linked, which is what rate theory would indicate [9]. The effect of stress on swelling was not explicitly shown in [9], so part of that analysis will be reproduced here with an explicit example to show how stress affects swelling. A simple rate theory construct is shown schematically for the simple case of a uniaxial stress in Figure 12. It is assumed that the kinetics are sink-dominated (temperatures 300 °C < T < 500 °C for steels and < 300 °C for Zr-alloys), and recombination can therefore be ignored. The normal dislocation bias (b), calculated following the expressions given by [43], is taken as 0.3. Following the methodology described by Woo [42], it is further assumed that the stress induces an additional interstitial bias (s) that is anisotropic, and (we will assume) half the value of the normal dislocation bias is, for example, 0.15, for a given stress.

Using the simple model shown in Figure 12, one can compute the swelling with and without an applied stress and examine the effect of varying stress, dislocation sink strength, and cavity sink strength. These results have been reported previously [9]. The calculated swelling rates as a function of the swelling for two dislocation densities $\left(\rho =10^{14}\;m^{-2}and\;\rho =10^{15}\;m^{-2}\right)$ are shown in Figure 13 together with the calculated value for D (the creep compliance), assuming the value of the stress bias (s = 0.15) is equivalent to 100 MPa. The main points to note from this simple model are that: (i) the swelling rate is dependent on the swelling and the dislocation density, i.e., the ratio of the unbiased dislocation and cavity sink strengths ($SS_{disl}\;and\;SS_{cav}$ respectively); (ii) in the absence of a stress (zero creep) the swelling is reduced; (iii) for stress free swelling (s = 0) the swelling rate is a maximum when $SS_{cav}=\sqrt{1+p}\cdot SS_{disl}$, where p is the normal dislocation bias, and is a more complicated dependence on p and s when s≠0; and (iv) the swelling creep compliance (D) is a function of the amount of swelling (cavity volume) and the network dislocation density. According to the simple model, both the creep rate and the swelling rate should be a function of the microstructure as it evolves during irradiation [10].

Not explicitly reported in [9] was the deviatoric component of the creep strain. Figure 14 shows the effect of increasing the stress bias parameter (s), which is proportional to the stress, on the linear strain from swelling and the deviatoric creep strain. Two cases were considered: (i) steady state dislocation density = 10^15^ m^−2^; and (ii) steady state dislocation density = 10^14^ m^−2^. These densities correspond with the steady state values for fuel pins irradiated in DFR at temperatures of about 400 °C and 600 °C for 20% cold worked 316 SS [33,34]. It is assumed that the microstructure is isotropic, and the dislocations are equally partitioned over the three directions. The creep and swelling rates are then calculated with and without an applied stress (assume an equivalent to 100 MPa), but as a function of an evolving cavity structure. For a cavity diameter of 20 nm, the calculated linear swelling rates ($\frac{swelling\;rate}{3}$) and the deviatoric creep strain rates as a function of the stress bias parameter (s), for dislocation densities of 10^14^ and 10^15^ m^−2^ and cavity densities corresponding with 0% and 20% swelling are shown in Figure 14. The total strain rate for a given dislocation density (10^14^ and 10^15^ m^−2^) is the sum of the deviatoric strain rates and the linear strain rates due to swelling. The total linear strain rate in the absence of swelling is independent of the dislocation density. There are two important points to note from this basic result: (i) the swelling is not independent of stress, as one might imagine; and (ii) the creep rate in the absence of swelling is independent of the dislocation density. This means that if there are only dislocation sinks during irradiation, then from the simple model shown in Figure 12, creep is dependent on the magnitude of the bias and not on the absolute number of dislocations; it is the ratio that is important, not the absolute number when there is only one biased sink to consider. The latter observation is consistent with the results shown in Figure 10. It is noteworthy that, for the data shown in Figure 10, the measured axial strain at zero stress, which is an indication of the linear swelling, is very small even at 60 dpa [56]. Similarly, for stress free swelling, it is the ratio of the dislocation to cavity sink strengths that is important in dictating the swelling rate, not the absolute numbers.

Anisotropic swelling [60], which can arise from a non-uniform distribution of dislocations, and any non-linear variation of swelling with stress complicates the analysis of swelling and irradiation creep strains [61,62,63,64]. Part of the complication in deriving an empirical expression that includes swelling and creep is the assumption that swelling and creep are independent and separable. Because swelling is dependent on diffusional mass transport, it must also be affected by that part of the creep that is also dependent on diffusional mass transport. If stress has any effect on the diffusion of point defects, the swelling cannot then be independent of stress [65], and can only really be described with a rate theory model that also includes the effect of stress on mass transport. Although it is expected that stress will affect diffusion anisotropy [42], there are no definitive data to determine exactly how stress affects either the diffusion of vacancies or self-interstitials during irradiation. For all practical purposes, the swelling data have therefore been processed assuming that swelling is independent of stress. In the model proposed by Ehrlich [20], the net diffusion of the interstitial point defect to dislocations governing the creep rate is proportional to the instantaneous swelling rate, provided the only sink strength that is varying is that of the cavities. It is the relative sink strengths of the dislocations and cavities that governs the partitioning of the point defects to both sink types [66].

In addition to considerations of rate theory, which depends on the relative sink strength of all sinks, the evolution of creep compliance (D) has a different relationship with the swelling depending on whether the swelling is dictated by simply increasing the number of cavities of constant diameter or vice-versa, and the swelling is determined by a fixed number density (established at low doses) and an increase in the mean cavity diameter. If the cavity number density and the dislocation density are constant, the swelling is dictated by cavity growth. The instantaneous swelling and creep rates, which are dependent on the relative sink strengths of cavities and dislocations, will then be proportional to $\sqrt[{3}]{S}$, where S is the swelling, because the sink strength is 4πrρ and the swelling is $\frac{4}{3}\pi r^{3}\rho$. Both the swelling and creep rates will be a non-linear function of the swelling, and thus dose, so it is important that one use the instantaneous swelling rate when determining what the creep rate should be. It is only when the cavities have a constant radius and the swelling evolves due to an increasing number density that linearity of the sink strength with swelling, and thus swelling with dose, can be expected. When the number density and cavity sizes are both increasing, the instantaneous swelling rate (proportional to the creep rate) will be a more complex function of dose.

In their analysis of swelling and creep data, Ukai and Ohtsuka [67] proposed that D varies non-linearly with swelling. In their analysis, they calculate an effective value for D that is averaged over a dose range up to the point when the swelling is measured. It is not necessarily that D varies, rather that the swelling rate varies, and by integrating the swelling over a given dose range they generate an effective value for D by essentially assuming a constant swelling rate. It is therefore not surprising that their effective D is a non-linear function of the swelling. They attribute the decline of the coefficient D with swelling to the partitioning of point defects between dislocations, voids, and precipitates. While their analysis is reasonable and a decreasing value of D as the swelling increases for a constant dislocation density is consistent with rate theory [9], it should be noted that the variation in D with swelling calculated by Ukai and Ohtsuka [67] is for an effective D derived when the swelling is integrated over a given dose range rather than on the actual value of D corresponding to the instantaneous swelling rate (Equation (1)), which is often assumed to be constant but is actually not, simply based on rate theory [9] (see Figure 13). To summarise, only if the swelling is dictated by an increase in the number density of cavities of constant diameter can the swelling rate be constant with dose for a given dislocation density. The creep rate is a function of the instantaneous swelling rate because they are coupled by the same process—diffusional mass transport.

For austenitic alloys, the irradiation creep that is dependent on the swelling rate must be controlled by diffusional mass transport. Bearing in mind that creep is a constant volume process, one cannot have a model for that aspect of irradiation creep without a model for swelling. Whereas it is reasonable to expect that the swelling-independent creep (*B*_0_) may be dominated by dislocation slip up until a steady-state is achieved (<1 dpa for Zr-alloys and <10 dpa for austenicic alloys), the swelling dependent part ($D\dot{S}$) is not, and one must therefore consider all possible strain producing mechanisms when modelling irradiation creep of austenitic alloys at high doses.

### 3.4. Effect of Grain Structure on Irradiation Creep

Grain structure can be important for irradiation creep when the grain dimensions are small, because a high density of grain boundaries will act as strong sinks for both gliding dislocations and point defects. For dislocation slip, small grains limit the distance a dislocation can travel before it is eliminated. One can therefore expect that if one observes high irradiation creep rates with larger grained material, then the dominant mechanism is likely to be slip. Conversely, if the opposite is true (higher creep for smaller grain dimensions) then it is less likely that slip is the main strain producing mechanism. For austenitic alloys the grain dimensions are often large (>5 μm) and are therefore unlikely to have any significant effect on creep strain. For certain Zr-alloys, cold-worked and stress-relief-annealed fuel cladding or Zr-2.5Nb pressure tubing, the grain dimensions are small enough (<1 μm) that they can limit the creep strain from slip and compete with network dislocations for point defects during irradiation. For Zr-2.5Nb pressure tubes, the grain structure is anisotropic, and most grains can be described as platelets whose mean dimensions are typically <0.6 μm, 2–6 μm, and >10–20 μm in directions corresponding to the radial, transverse (hoop), and longitudinal direction of the tubes, respectively [46,47]. To study the effect of microstructure variables on irradiation creep, one needs a model to normalise for operating conditions. Two types of models have been applied in two different studies, one empirical [22] and one semi-empirical [25]. In both cases the creep was shown to be strongly dependent on variations in grain structure [49,50]. Even though front-to-back variations in texture and grain structure are subsumed into the temperature term in [22] and the front-to-back variation in grain structure was accounted for by the K_4_(*x*) term in [25], the normalisation models for operating conditions allow for the assessment of the relative creep rates of different pressure tubes at the same axial locations [49,50]. Using grain dimensions, crystallographic texture and dislocation density, a rate theory model was developed that accounted for the relative irradiation creep behaviour of different tubes based on differences in microstructure [49,50]. In the earlier model, the strain was calculated based on the orientation of various sinks. Assuming a sharp texture where all grains were oriented so that their basal poles were parallel with the transverse direction, the model showed that grain dimensions were important. Recently, a more refined model has been developed that associates grain shape with texture, as illustrated in Figure 15 [10].

It is assumed that the sink orientation is biased for interstitial absorption following the DAD mechanism according to the bias parameter (p) indicating preferential diffusion in the basal plane [68]. The dislocation bias for interstitials is denoted by the parameter, b, based on the elastic size-effect interaction between dislocations and point defects [43,69]. The dislocation bias will also be modified by the DAD bias due to the line orientation [70], but for the sake of this treatment we will assume that any additional effects due to anisotropic diffusion and line orientation are subsumed into the dislocation bias term (b). The effect of stress can be incorporated as a bias parameter (s) resulting from elasto-diffusion [42].

If one is operating in a temperature regime where recombination and vacancy emission are insignificant, one can represent the net flux of interstitials and vacancies to sinks, resulting in strain in the radial (*R*), transverse (*T*), and longitudinal (*L*) directions of a given grain of dimensions d_R,T,L_, by the following expressions:(20)$$J_{R}=\sum_{R}k_{i}^{2}\;\cdot D_{i}C_{i}-\sum_{R}k_{v}^{2}\cdot D_{v}C_{v}$$
(21)$$J_{T}=\sum_{T}k_{i}^{2}\;\cdot D_{i}C_{i}-\sum_{T}k_{v}^{2}\cdot D_{v}C_{v}$$
(22)$$J_{L}=\sum_{L}k_{i}^{2}\;\cdot D_{i}C_{i}-\sum_{L}k_{v}^{2}\cdot D_{v}C_{v}$$
(23)$$D_{i}C_{i}=\frac{\phi }{\sum_{L}k_{i}^{2}+\sum_{T}k_{i}^{2}+\sum_{R}k_{i}^{2}}D_{v}C_{v}=\frac{\phi }{\sum_{L}k_{v}^{2}+\sum_{T}k_{v}^{2}+\sum_{R}k_{v}^{2}\;}$$
where *ϕ* is the FMD production rate, $D_{i}$ and $D_{v}$ are the interstitial and vacancy diffusion coefficients, and $C_{i}$ and $C_{v}$ are the interstitial and vacancy steady state concentrations. If the sink strengths internal to the grains are given by $k_{i,v}{}^{2}$ then the sink strength for grain boundaries perpendicular to the directions *R*, *T* and *L* are given by [71],
(24)$${\left(k_{i,v}^{gb}\right)}^{2}=\frac{2\cdot k_{i,v}}{d_{R,T,L}}$$

The sink strengths of the a- and c-type network dislocations are determined from the dislocation densities, ρ_a_ and ρ_c_. The sink strength of the dislocation loops (assumed neutral) is given by ρ_N_. The total internal sink strength dictates the grain boundary sink strength [71].

For a grain that has the c-axis parallel with the transverse direction, the sink strengths used to compute the strains in the radial (*R*), transverse (*T*) and longitudinal (*L*) directions are, respectively:(25)$$\mathrm{Radial}:\;\sum k_{i}^{2}=\left(1+p\right){\left(k_{i}^{gb}\right)}^{2}+\left(1+b\right)\frac{\rho_{a}}{2}+\frac{\rho_{N}}{3}{{;}_{\;}}_{\;}\sum k_{v}^{2}={\left(k_{v}^{gb}\right)}^{2}+\frac{\rho_{a}}{2}+\frac{\rho_{N}}{3}$$
(26)$$\mathrm{Transverse}:\;\sum k_{i}^{2}=\left(1+2s\right){\left(k_{i}^{gb}\right)}^{2}+\left(1+2s\right)\rho_{c}+\frac{\rho_{N}}{3};\sum k_{v}^{2}={\left(k_{v}^{gb}\right)}^{2}+\rho_{cl}+\frac{\rho_{N}}{3}$$
(27)$$\mathrm{Longitudinal}:\;\sum k_{i}^{2}=\left(1+p+s\right){\left(k_{i}^{gb}\right)}^{2}+\left(1+b+s\right)\frac{\rho_{a}}{2}+\frac{\rho_{N}}{3};\;\sum k_{v}^{2}={\left(k_{v}^{gb}\right)}^{2}+\frac{\rho_{a}}{2}+\frac{\rho_{N}}{3}$$
where $\rho_{cl}=\rho_{c}\left(1+c\cdot dpa\right)$ represents the c-dislocations, including a c-component dislocation loop structure evolving by helical climb (see Figure 4 and Figure 5). For cold-worked Zr-2.5Nb pressure tubing, an approximate estimate for c is 0.04/fmd, which is about 0.004/primary dpa, based on the line broadening of cold-worked Zircaloy-2 pressure tube material [53]. The effect of allowing the c-component vacancy loop sink strength to evolve by helical climb in a model such as this is a decreasing rate of diametral creep with increasing dose [10]. This decreasing rate of creep with increasing dose (because of the c-loop evolution) is shown by the data circled in Figure 6.

For a grain that has the c-axis parallel with the radial direction, the sink strengths used to compute the strains in the radial (*R*), transverse (*T*) and longitudinal (*L*) directions are, respectively:(28)$$\mathrm{Radial}:\;\sum k_{i}^{2}={\left(k_{i}^{gb}\right)}^{2}+\rho_{c}+\frac{\rho_{N}}{3}{{\;}_{;}}_{\;}\sum k_{v}^{2}={\left(k_{v}^{gb}\right)}^{2}+\rho_{cl}+\frac{\rho_{N}}{3}$$
(29)$$\mathrm{Transverse}:\;\sum k_{i}^{2}=\left(1+p+2s\right)\cdot {\left(k_{i}^{gb}\right)}^{2}+\left(1+b+2s\right)\cdot \frac{\rho_{a}}{2}+\frac{\rho_{N}}{3};\;\sum k_{v}^{2}={\left(k_{v}^{gb}\right)}^{2}+\frac{\rho_{a}}{2}+\frac{\rho_{N}}{3}$$
(30)$$\mathrm{Longitudinal}:\;\sum k_{i}^{2}=\left(1+p+s\right)\cdot {\left(k_{i}^{gb}\right)}^{2}+\left(1+b+s\right)\cdot \frac{\rho_{a}}{2}+\frac{\rho_{N}}{3};\;\sum k_{v}^{2}={\left(k_{v}^{gb}\right)}^{2}+\frac{\rho_{a}}{2}+\frac{\rho_{N}}{3}$$

The grain boundary sink strengths are assumed to be determined from the internal sink structure without the biases based on diffusion anisotropy, i.e., independent of intrinsic bias (p) and stress-induced bias (s) because these biases are also applied to the boundaries in the balance equations:(31)$$\mathrm{Radial}:{\;}_{\;}{\left(k_{i}^{gb}\right)}^{2}={\left(k_{v}^{gb}\right)}^{2}=\frac{2\cdot \sqrt{\left(1+b\right)\cdot \;\rho_{a}+\rho_{cl}+\rho_{N}\;}}{d_{R}}$$
(32)$$\mathrm{Transverse}\text{:}{\;}_{\;}{\left(k_{i}^{gb}\right)}^{2}={\left(k_{v}^{gb}\right)}^{2}=\frac{2\cdot \sqrt{\left(1+b\right)\cdot \;\rho_{a}+\rho_{cl}+\rho_{N}}}{d_{T}}$$
(33)$$\mathrm{Longitudinal}:\;{\left(k_{i}^{gb}\right)}^{2}={\left(k_{v}^{gb}\right)}^{2}=\frac{2\cdot \sqrt{\left(1+b\right)\cdot \;\rho_{a}+\rho_{cl}+\rho_{N}}}{d_{L}}$$

The main value of developing a mechanistic model of this type is to give some insight into the possible mechanisms that apply to irradiation creep. Rate-theory has proven to be important in explaining the swelling dependent part of the irradiation creep of austenitic alloys, and it could equally apply to the swelling independent creep of Zr-alloys. In the latter case, irradiation creep at high doses can be envisaged as a continuation of the shift from dislocation slip dominating the irradiation creep behaviour to something else, in this case mass transport (see Figure 6).

Exact agreement with the experimental data are difficult to attain because of the large uncertainty in the value of many of the underlying parameters that feed in to a rate theory model such as the one described here, but also because of material variability and the experimental difficulties in obtaining measurements of small strains over long periods. However, a microstructure model of this type can be useful in comparing the behaviour of different materials where the microstructures have been characterised. To better understand the irradiation creep behaviour of Zr-2.5Nb pressure tubing, it is fortunate that one reactor operator allowed some specially fabricated pressure tubes to be installed in one reactor. These tubes have been periodically monitored and gauged over a 30-year period together with other tubes fabricated following a standard route. The tubes in question were fabricated to meet the required operating specifications (e.g., strength and corrosion) but were tailored in such a way as to minimise elongation during irradiation by reducing the dislocation density, which was perceived at that time as the cause of higher longitudinal creep rates for tubing. To maintain good strength properties, the tubes were made with slightly small grains. The net effect was the production of tubes (designated as route 1) that met the required physical property specifications but had a slightly different microstructure (smaller grains and lower dislocation density) compared with the standard processing route [72]. At this point in time, the gauging of these tubes has shown that they do exhibit lower elongation, and in that sense the project was successful. More importantly, it has provided data on the long term irradiation creep behaviour of pressure tubing that can be used in the development of mechanistic models involving microstructure.

Detailed measurements of microstructures from offcuts taken from each end of the three route 1 (RT1) tubes and three standard pressure tubes irradiated in the same reactor are shown in Table 1.

These data have been incorporated in the rate theory model described above. Because dislocation densities vary significantly in different samples, measurements of a single specimen cannot be deemed representative of the whole tube. For this reason, the dislocation densities were assumed to be 4 × 10^14^ m^−2^ for a-type (ρ_a_) and 1 × 10^14^ m^−2^ for c-type (ρ_c_) in the standard pressure tubes to be consistent with historic averages from creep capsules and pressure tubing [26,45,46,47]. The values for the RT1 tubes were chosen to be slightly small, 3 × 10^14^ m^−2^ for a-type (ρ_a_) and 0.5 × 10^14^ m^−2^ for c-type (ρ_c_), to be consistent with the relative measurements shown in Table 1.

The model parameters for dislocation bias (b) and intrinsic diffusion bias (p) were chosen to be 0.4 and 0.3, respectively, consistent with theoretical modelling [43,68]. The stress bias (s), attributed to elasto-diffusion [42] has been determined from the most reasonable fits to creep capsule data. Figure 16 shows the model output as a function of s obtained by applying the reported mean texture parameters and mean grain thickness (with an assumed aspect ratio of 5) for creep capsules irradiated in NRU [26]. The value of s is a function of stress that gives the most reasonable agreement with the data with a value of $s≅0.003\cdot \sigma_{H}$, where $\sigma_{H}$ is the hoop stress. It is apparent that the prediction would give a large negative strain at zero stress (corresponding with irradiation growth) whereas the data (obtained at low doses) indicates a degree of curvature so that the intercept at zero stress is likely to be small. This non-linear (or offset) trend for the compliance is also exhibited by austenitic alloys [8,57]. The offset shown by the model described here is because it is designed to capture the creep rates of pressure tubing after large operating times, i.e., when the c-component dislocation density is increasing constantly. The evolution of the c-component loop structure reduces the creep rate [10]. In a stress-free environment, the irradiation growth is known to be accelerating [18]. The growth rate is accelerating constantly, so that at after a dose of about 15 × 10^25^ n·m^−2^ (~25 dpa), the measured growth rates can be as high as 10 × 10^−29^ m^2^.m^−1^ (see Figure 5.4.1 in [18]), and will be higher as the dose is increased further. For this reason, a large negative intercept for the strain rate is perhaps not too surprising because the model assumes a constant rate of c-component loop evolution corresponding to long operating times, and the negative intercept will increase as the c-loop structure evolves, lowering the creep rate. Furthermore, the estimates for the c-component dislocation densities are subject to sample variability, having a wide range of measured values for different creep capsules [26]. The non-linearity with stress is expected at low stresses because the stress bias is competing with the intrinsic sink bias (from anisotropic diffusion), but the stress bias will dominate as the stress increases.

Lastly, the a-type dislocation loop density must be estimated. The a-type loops, which can be regarded as neutral sinks, are important in aiding the recombination of interstitials and vacancies. When the a-type dislocation loop density is high, they absorb a large fraction of the available point defects of both signs. Prism plane line broadening measurements for one pressure tube having a neutron flux profile similar to that shown in Figure 1b have been reported previously [47], and selected data are shown in Figure 17. These data are fitted with a simple empirical model based on temperature, flux and fluence [74], which is also shown in Figure 17. Note that the data at any one axial location show a spread as a function of clock position arising from a circumferential variation in temperature as the pressure tubes expand as noted in [46,47,75], and this is because of the non-unform heating in the fuel channel because of the coolant flow-bypass of the fuel bundles (see inset in Figure 1a).

For the case of this model, it is assumed that the factor $\rho_{N}$ is given by,
(34)$$\rho_{N}\left(x\right)=\phi \left(x\right)\cdot \left(192\cdot \rho_{N}^{s}-46\cdot x\right)+\rho_{a}$$

This simple empirical relationship with flux and temperature (related in this case to the axial location relative to the inlet, *x*) gives the hypothetical profiles shown in Figure 18 for the cases being considered here. Because the dislocation loop density decreases with increasing temperature (see Figure 7 and Figure 16), this means that higher creep rates are expected at higher irradiation temperatures, i.e. towards the outlet of the fuel channel. This will be true whether one is considering the effect of the dislocation loops in hindering glide or as a competing neutral sink for point defects. The dislocation density within the operating pressure tube was estimated from the variation in line broadening along the pressure tube [46,47] that approximately matches the neutron flux profile, but is modified by the temperature (see Figure 18), and is scaled to give reasonable agreement with recent quantitative measurements on a-type dislocation loop densities in irradiated pressure tubing [75].

The results of applying the rate theory model to the pressure tubes listed in Table 1 are shown in Figure 19, assuming a stress bias factor of $s=0.0028\cdot \sigma_{H}$ that, for 125 MPa, corresponds with a value of the stress bias of s = 0.35. With the added constraint of having the c-loop multiplier for c that is 0.2/fmd to maintain reasonable agreement with the elongation rates, there is also reasonable agreement with measurements of diametral creep rates. The measurements apply to diametral strains at different times in the range between about 80,000 h and 200,000 h of operation, except for tube C, whose rates at different locations were determined from available measurements at 36,000 h and 78,000 h of operation. The corresponding predicted and measured elongation rates are shown in Table 2. Reasonable agreement is achieved without too much adjustment, while using a stress bias that best conforms to the creep capsule data. A better fit is obtained by choosing a stress bias of s = 0.29 and a c-loop multiplier for c that is 0.04/fmd, as shown in Figure 20. The corresponding predicted and measured elongation rates are shown in Table 3.

The most important feature of these results is not necessarily the agreement with the pressure tube creep strain rates, which can be adjusted depending on the choice of key model parameters, but rather the fact that there is consistent agreement in the relative difference in creep strain rates for the two groups of tubes given the variation in the microstructure parameters only. Using a model with the same fundamental physical constants, the relative behaviour of two groups of tubes can be reproduced by inputting their individual microstructure parameters without changing anything else. This agreement provides some support for the notion that diffusional mass transport is the dominant mechanism accounting for the irradiation creep of Zr-2.5Nb pressure tubes at temperatures < 300 °C and for stresses < 150 MPa. For Zr-2.5Nb pressure tubing, irradiation creep is very sensitive to the grain structure to the extent that tubes containing small grains that are more equiaxed in the radial-transverse directions exhibit the highest diametral creep but the lowest elongation.

## 4. Discussion

Creep in a nuclear reactor core is different from that out of the core because of the radiation damage, displaced (interstitial) atoms, and residual vacancies resulting in elevated steady-state point defect concentrations, which are dependent on the point defect production rate and the sink structure, of the order of 10^−7^ for vacancies and 10^−18^ for interstitials in an austenitic alloy in a power reactor at 300 °C. The steady state point defect concentrations during irradiation exceed the thermal equilibrium concentrations at the same temperature by many orders of magnitude, which are typically of the order of 10^−12^ and 10^−26^ for vacancies and interstitials respectively. The radiation-induced point defects can either recombine with each other, form new clusters, or migrate to existing sinks. The point defect clusters inhibit dislocation glide, leading to a drastic reduction in the creep rate at low doses as the dislocation loop structure evolves. SICG may still be a mechanism that contributes to the creep strain, even at high doses.

Thermally, at low stresses, dislocations can by-pass any barriers to slip by climb from vacancy absorption. In an irradiation environment, the climb and glide process is still applicable, but is more complicated because of the elastic interactions resulting in the preferential diffusion of interstitials to dislocations that may also be influenced by the applied stress [42,43]. It is therefore expected that the net interstitial flux to dislocations will govern the climb rate and thus the creep rate from dislocation glide when the barrier density is high. Whereas the rate-determining step for dislocation glide in a heavily irradiated material is climb, governed by point defect diffusion, the magnitude and direction of the creep is assumed in some models to come mostly from the slip [18,24,25] as opposed to the climb. For austenitic steels, the separation of irradiation creep by SICG (B_0_) and diffusional mass transport (D) is well understood [20]. The diffusional mass transport term (D) is generally not considered important until swelling commences [5,6], but the B_0_ compliance term still has some dependence on mass transport, and there is overlap of the two mechanisms at high doses in the swelling regime. Whereas diffusional mass transport clearly has an important role in irradiation creep at all doses, the extent to which the mass transport itself is the main strain producing mechanism, or whether dislocation slip aided by diffusional mass transport is the main mechanism, is still subject to debate. There are therefore two possible avenues for irradiation creep that are dependent on the rate of production of freely migrating point defects, as illustrated in Figure 21.

The mechanistic modelling of irradiation creep by slip and diffusional mass transport are complex and are outlined in the supplementary material for the case of Zr-2.5Nb pressure tubing, which is the focus of the modelling outlined in Section 3.4. As sink strengths for point defects are central to rate theory, the typical microstructures for cold-worked pressure tubes are first described and illustrated in Section S2 [8,10,50,72,73], Appendix A. To make a case for diffusional mass transport one must understand the strengths and shortcomings of dislocation slip. The role of dislocation slip is discussed in Section S3; firstly with respect to the potential effect of gliding prismatic dislocation loops on creep [8,76,77,78], Figure 5, and then with respect to the glide by network dislocations in a polycrystalline material [5,6,18,24,25,26,48,49,70,79,80,81,82,83,84,85,86,87], Appendix A. Finally, the fundamentals of rate theory are described by formulating a set of simple balance equations for dislocations [9,43,69,71,88,89], Appendix A, and then for grain boundaries [10,50], Appendix A.

Based on the evidence presented in this paper, the effect of irradiation is two-fold: (i) firstly, in creating radiation damage clusters that are barriers to slip and shut down conventional SICG; and (ii) in the enhancement of mass transport, not just from the elevation of vacancy concentrations, but also because of the higher concentrations of self-interstitial atoms. Given that there is weak dependence of irradiation creep on dislocation density [18,49,56], and given that rate theory models of mass transport can predict the relative behaviour of materials with different grain structures, one is drawn to the conclusion that diffusional mass transport is likely a dominant mechanism in certain ranges of temperature and stress. For Zr-alloys, when the temperature is high (>350 °C) so that radiation damage clustering is low, and/or the stress is high (>150 MPa), so that dislocation glide is activated more efficiently, one can expect dislocation slip to dominate. In austenitic alloys, because of the high vacancy migration energy at very low temperatures (<100 °C) dislocation slip also appears to be the dominant creep mechanism because there are no loops to inhibit glide while the point defect concentrations are still elevated relative to thermal levels. Rate theory calculations show that, for reactor core conditions, the net fluxes of self-interstitials to dislocations are many orders of magnitude larger than the fluxes of thermally-induced vacancies at temperatures < 500 °C, this being the basis for enhanced creep by dislocation glide during irradiation. However, radiation damage in the form of dislocation loops inhibits creep, and it is often not clear whether the enhancement of climb due to a large net interstitial flux can make up for the suppression of glide by point defect clusters at reactor operating temperatures. At temperatures > 500 °C, the thermally-induced vacancy flux exceeds the net irradiation-induced interstitial flux, i.e., the difference between interstitial and vacancy fluxes. At these elevated temperatures, thermal creep mechanisms will be dominant, provided that there is also a low density of point defect clusters, whether in the form of dislocation loops or cavities.

At high temperatures and moderate stresses, the irradiation creep of austenitic alloys is dependent not just on the flux of interstitial point defects to dislocations, which is the manifestation of swelling, but rather the anisotropy induced by a stress. The insensitivity of irradiation creep to dislocation density at reactor operating temperatures suggests that mass transport rather than SICG is the dominant mechanism under these conditions.

Bearing in mind the evidence to support diffusional mass transport as the dominant creep mechanism at low stresses, one can consider that irradiation creep from diffusional mass transport is merely an expansion of Nabarro-Herring creep to lower temperatures and higher stresses because of irradiation, as illustrated in Figure 22 [58]. The map chosen from [58] is for pure silver because it shows the main creep mechanisms that apply to many other metals such as iron and nickel but at different stress levels. The temperature range is extended to lower temperatures because the steady-state point defect concentrations of vacancies and, perhaps more importantly, interstitials, are elevated. Whereas in the thermal case the Nabarro-Herring creep mechanism involves vacancy diffusion only and is therefore important at high temperatures, where vacancies are both abundant and mobile, the point defect fluxes are also high at low temperatures when the material is being irradiated, and creep from mass transport is therefore a viable proposition. Diffusional mass transport may also be more important at higher stresses because dislocation slip is inhibited by irradiation that increases the yield strength of the material, making it harder for dislocation glide to occur.

The issue of whether irradiation creep is dominated by diffusional mass transport or dislocation slip (albeit aided by dislocation climb) is important with respect to creep ductility. Consider a simple tensile test. The accelerating engineering creep rate in the tertiary stage of creep [8,10] is, in part, a function of the increase in true stress that occurs as a specimen’s cross section is decreased because of the creep. When dislocation slip is a dominant part of the creep mechanism, incompatibility stresses can arise between a plastically deforming matrix and inclusions that may be present in the material. Such stresses could lead to a stress state at the interface with the inclusion that induces micro-void formation. Subsequent micro-void growth and coalescence further reduces the effective cross section of the material and contributes to accelerating tertiary creep and eventual failure. When irradiation creep is dominated by diffusional mass transport, any incompatibilities between different phases will be reduced by the very nature of the diffusional creep process, i.e., stress relief by diffusion will reduce the stresses that could lead to micro-crack/void formation at the interfaces.

We now turn to the dominance of dislocation slip or diffusional mass transport in irradiation creep. Whereas circumstantial evidence has been presented to indicate which mechanism dominates at what temperature, stress and dose range, it should be possible to conduct definitive experiments to resolve the issue. Irradiation creep tests on material with varying degrees of cold-work starting from zero up to 30% could be conducted using: (i) large-grained material (>10 μm diameter); and (ii) material with small grain dimensions (<1 μm) in at least one direction. Testing should be conducted with a range of stresses and temperatures relevant to power reactor operation.

## 5. Conclusions

(i) Irradiation creep is a complex process that involves the elevated concentration of freely-migrating point defects produced by irradiation that promotes the climb of network dislocations and dislocation loops. Climb itself contributes to the strain in a given direction under the action of a stress. This can either be considered simply in terms of work done or in terms of the diffusional drift of point defects so that the climb of dislocations of a given Burgers’ vector is enhanced by elasto-diffusion or SIPA. For enhanced climb and glide under the action of a stress (SICG), the hardening effect of the irradiation on the microstructure is important, effectively reducing the creep per unit dose in the early stages of irradiation until the loop density reaches a constant state and the creep from dislocations (from either climb or glide) reaches a steady-state condition.

(ii) It is difficult to know how much of the creep strain comes directly from climb and how much comes from glide. In this context, the glide component must eventually be exhausted, either because the dislocations have travelled far enough to leave the crystal, or they are blocked by additional barriers to slip, i.e., cavities, that evolve over a longer period compared with the dislocation loop structure. Irradiation creep at low temperatures (<350 °C for Zr-alloys and 100 °C < T < 500 °C for austenitic alloys), and low stresses (<150 MPa for Zr-alloys and <200 MPa for austenitic alloys) is deemed to be a direct result of diffusional mass transport.

(iii) The irradiation creep processes that affect steady-state irradiation creep of austenitic alloys or Zr-alloys can either be some form of enhanced glide (SICG) or enhanced climb (SIPA), or simply the effect of stress on the direction of diffusion (elasto-diffusion). The accumulation of irradiation damage at low doses effectively blocks and inhibits the glide of network dislocations.

(iv) In austenitic alloys at very low irradiation temperatures (<100 °C) and low doses, irradiation creep is dependent on the dislocation glide that is uninhibited by point defect clustering and is thus faster than creep at higher temperatures (300 °C < T < 500 °C), where point defect clustering is more prevalent.

(v) In austenitic alloys in the void swelling regime, irradiation creep is dependent on diffusional mass transport because of the inter-relation with swelling. Swelling is dictated by the sink strength (size and number density) of the neutral sinks (cavities) that play an important role in controlling the partitioning of the point defects between the dislocations and the cavities. For a given neutron flux in the absence of helium, that part of irradiation creep contributing to the creep compliance (D) will diminish at temperatures > 500 °C because of the decrease in swelling. In this temperature range, the thermal vacancy flux to dislocations exceeds the net interstitial flux, and creep due to dislocation slip (conventional thermal creep) will increase, provided that there are sufficient gliding dislocations. Because diffusional mass transport depends on a balance of fluxes of interstitials and vacancies to different sinks, the swelling creep compliance (D) itself is not independent of swelling.

(vi) Irradiation creep is intimately dependent on the microstructure, whether it be the dislocation loops and cavities produced during irradiation, or the as-fabricated microstructure. In this respect, irradiation creep is insensitive to dislocation density for both Zr-alloys and austenitic alloys but is very sensitive to the grain structure (size and shape) for Zr-alloys.

## Figures and Tables

**Figure 1 materials-16-02287-f001:**
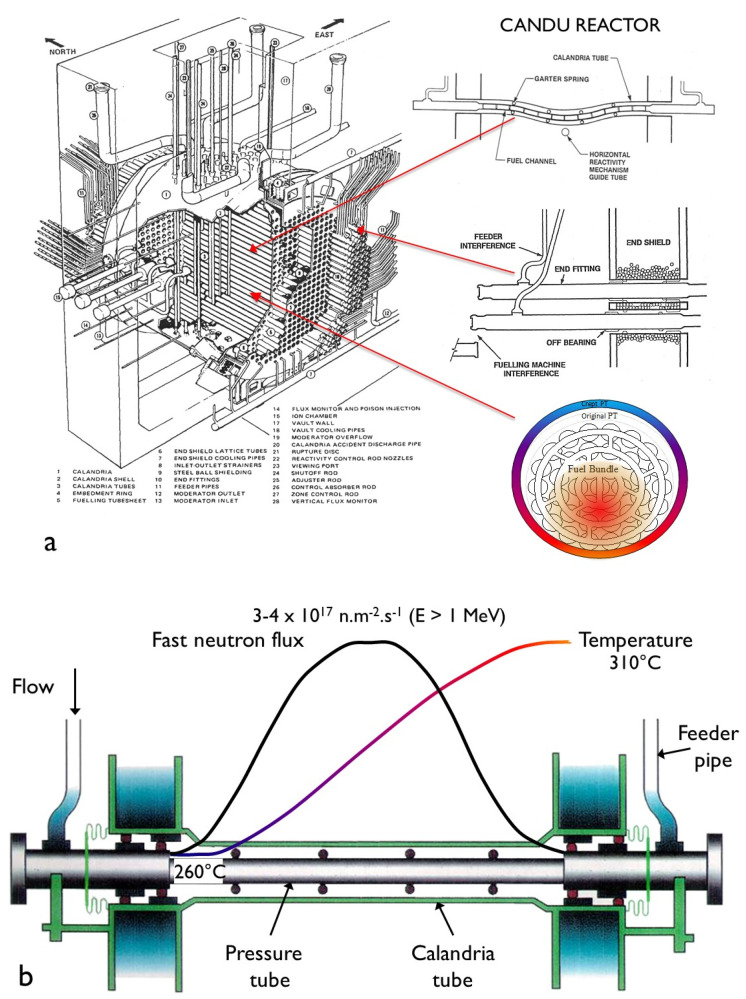
(**a**) schematic showing a CANDU reactor comprised of pressure tubes containing the fuel and coolant that also shows how the effect of pressure tube deformation (sag, elongation and diametral expansion) can interfere with reactor operation; (**b**) schematic showing a single fuel channel illustrating how the temperature and flux varies along the channel.

**Figure 2 materials-16-02287-f002:**
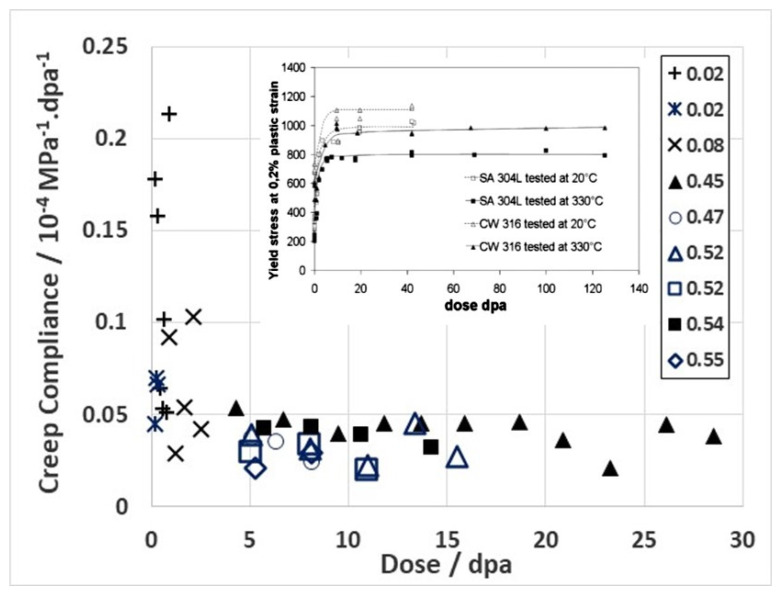
Creep as a function of dose rate for 316 SS springs irradiated in Dounreay fast reactors at temperatures between 520 K–633 K (247–360 °C). The inset shows yield stress as a function of dpa for 316 SS irradiated in the BOR 60 reactor at 330 °C [38]. The legend indicates different displacement damage rates in units of 10^−7^ dpa·s^−1^, the lowest corresponding to the lowest dose data.

**Figure 3 materials-16-02287-f003:**
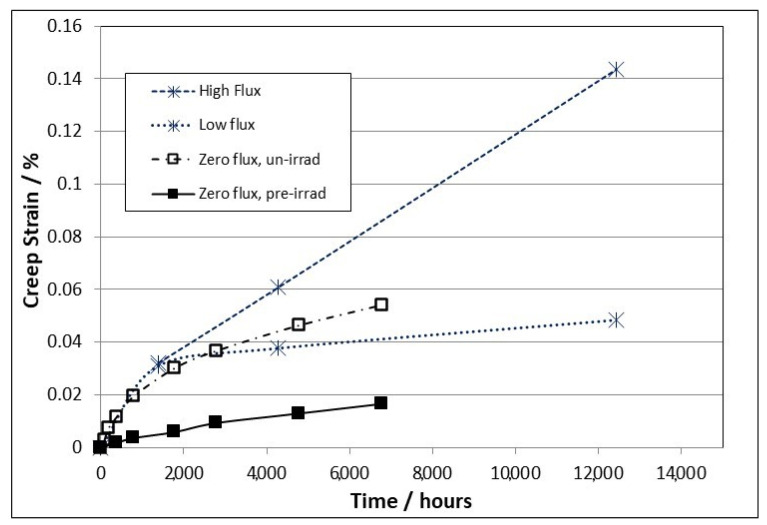
Out-reactor creep of pressurised creep capsules. The temperature is 280 °C and the hoop stress is about 125 MPa in each case.

**Figure 4 materials-16-02287-f004:**
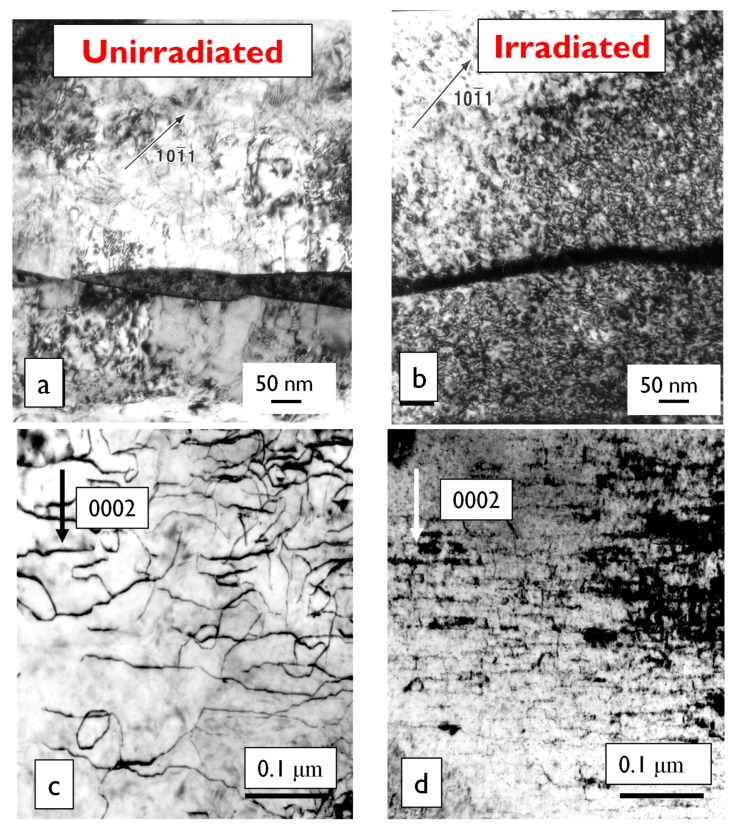
Evolution of a-type dislocation structure (**a**,**b**) and c-component dislocation structure (**c**,**d**) in Zr-2.5Nb pressure tubing after irradiation to a fluence of 1.1 × 10^26^ n·m^−2^ at 270 °C.

**Figure 5 materials-16-02287-f005:**
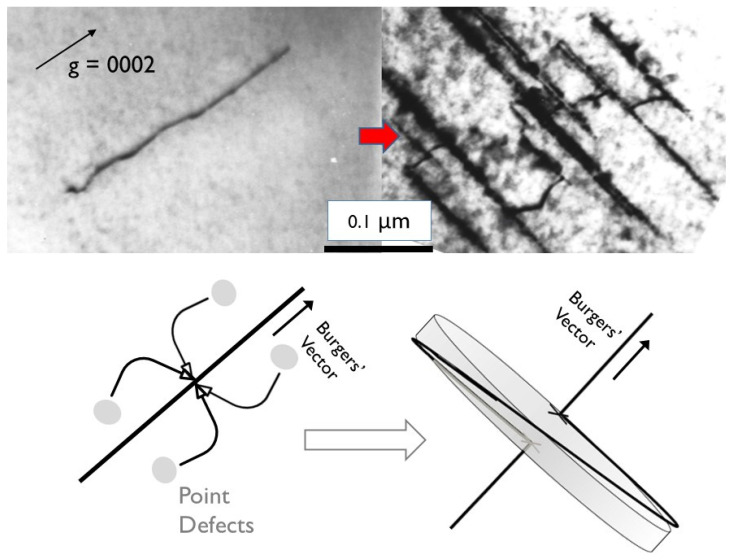
Micrographs and schematic of the helical climb of a straight c+a screw dislocation in Zr during electron irradiation at about 400 °C to a dose > 10 dpa.

**Figure 6 materials-16-02287-f006:**
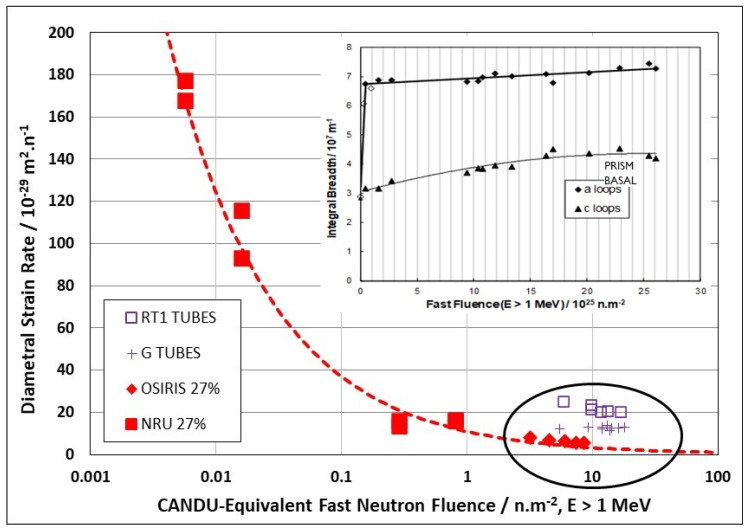
Irradiation creep rates (per unit fluence) as a function of fast neutron fluence for Zr-2.5Nb having similar textures but made by different manufacturing routes and therefore having different dislocation and grain structures. The temperature and hoop stress are about 280 °C and 125 MPa, respectively. Modified from [10].

**Figure 7 materials-16-02287-f007:**
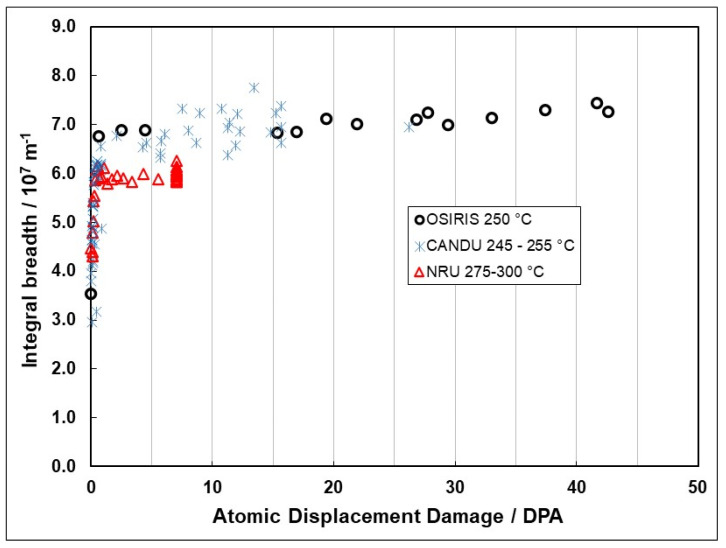
Prism plane line broadening (integral breadth) measured from Zr-2.5Nb pressure tube samples irradiated in OSIRIS and the NRU reactor. Also shown are data from >20 different different tubes after service in a CANDU reactor.

**Figure 8 materials-16-02287-f008:**
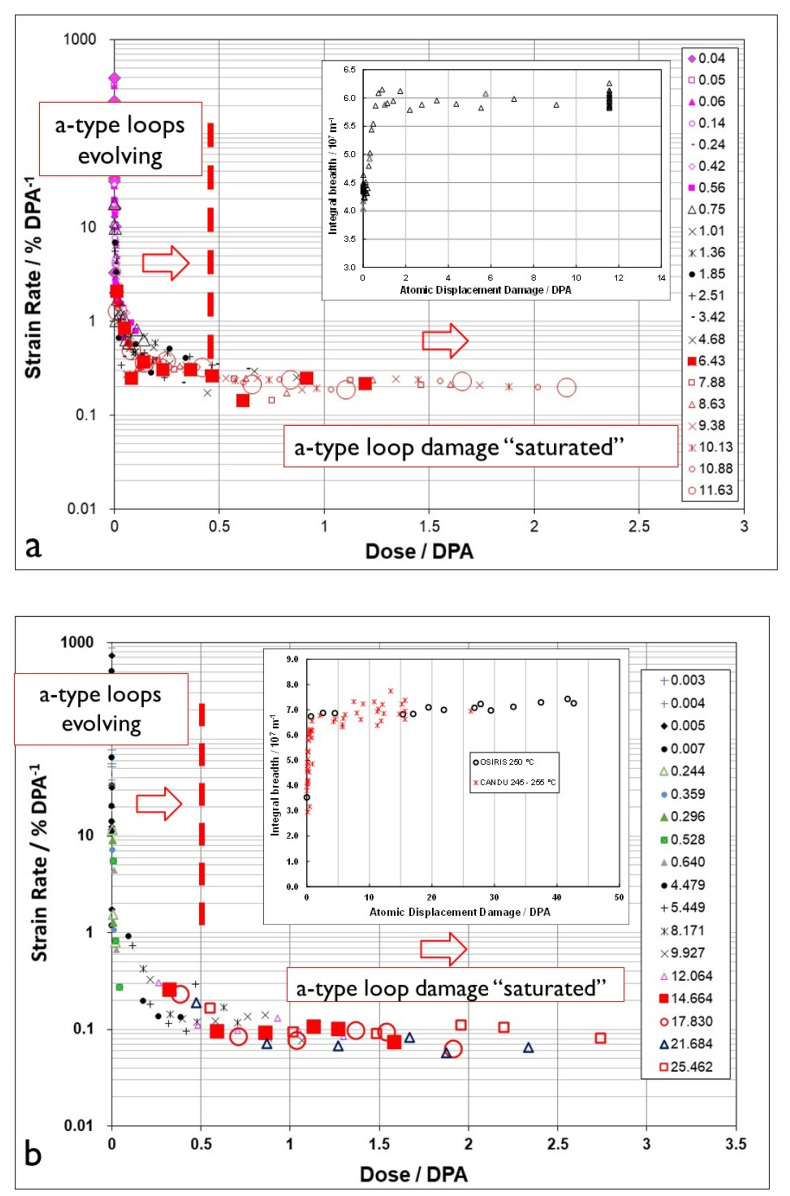
Irradiation creep rate (strain per unit dpa) as a function of dpa during irradiation in NRU at about 300 °C (**a**) and 275 °C (**b**). The legend indicates different displacement damage rates in units of 10^−10^ dpa·s^−1^, the lowest corresponding to the lowest dose data. The relationship with the evolution of the dislocation loop density is illustrated by the inset in (**a**) showing the line broadening measured from the ex-service material at 275–300 °C. The inset shown in (**b**) is not for the same material but is the low temperature data (250 °C) shown in Figure 7. The red arrows denote the fluence range for two regimes of the radiation damage (dislocation loop) evolution before and after the vertical red dashed line: (i) increasing density, (ii) constant density (saturated).

**Figure 9 materials-16-02287-f009:**
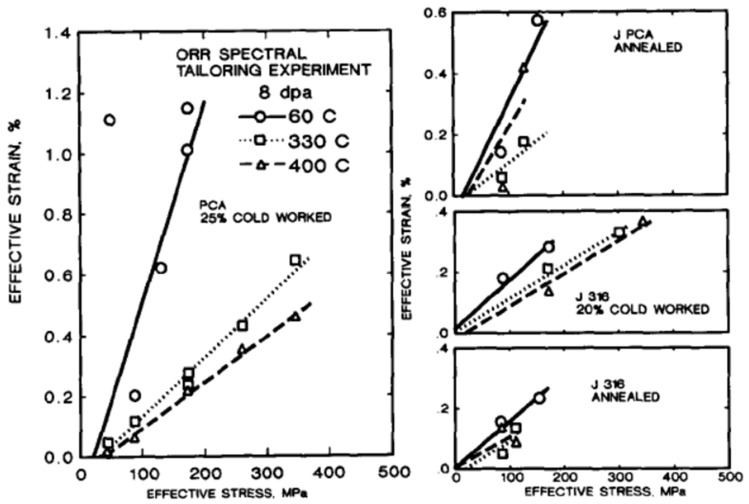
Total effective creep strain as a function of effective stress at a dose of 8 dpa for primary candidate alloy (PCA) and AISI 316 SS irradiated in an ORR spectral tailoring experiment. The high creep strains at a temperature of 60 °C are evident. Reprinted/adapted with permission from [55], 1990, Elsevier.

**Figure 10 materials-16-02287-f010:**
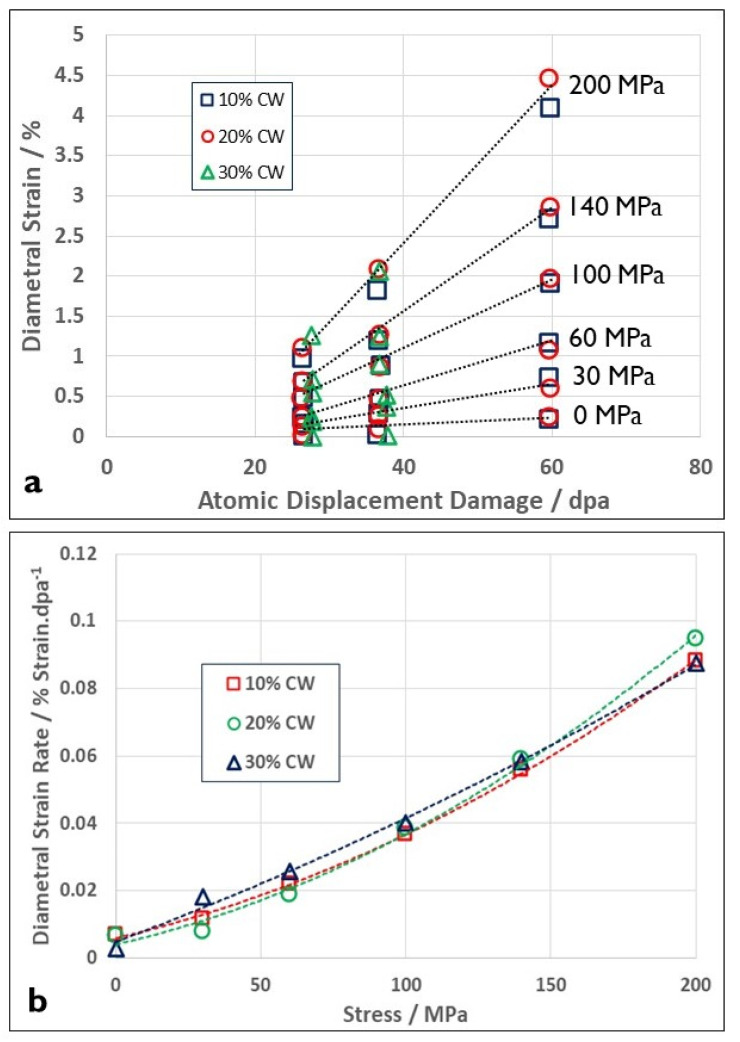
Deformation of 316 stainless steels induced by irradiation creep and swelling of pressurized creep capsules at 400 °C for three cold-work levels: (**a**) diametral strain versus dose (dpa); (**b**) axial strain rate versus stress (MPa).

**Figure 11 materials-16-02287-f011:**
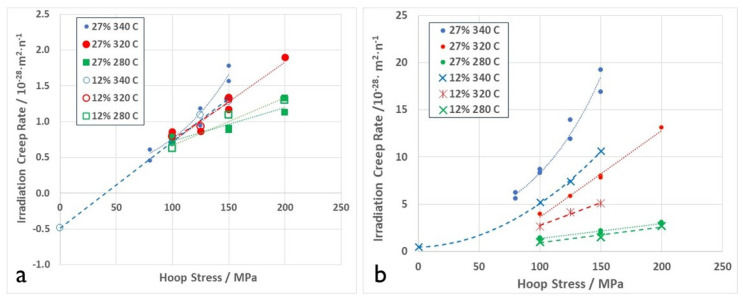
Creep rates (per unit fluence) as a function of hoop stress for Zr-2.5Nb 12% and 27% cold-worked creep capsules irradiated in NRU at different temperatures: (**a**) axial creep; (**b**) diametral creep.

**Figure 12 materials-16-02287-f012:**
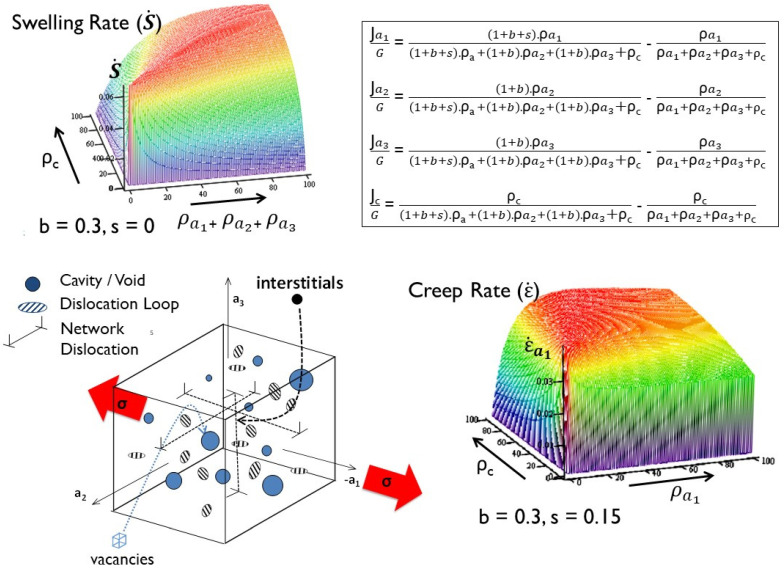
Simple rate theory formulation for calculating steady-state swelling and uniaxial creep rates as a function of the microstructure evolution. In this example the dislocation density is chosen to be 10^15^ m^−2^, which is the expected density from residual cold-work and radiation damage (dislocation loops) in austenitic alloys at temperatures > 300 °C. The creep includes the strain from swelling.

**Figure 13 materials-16-02287-f013:**
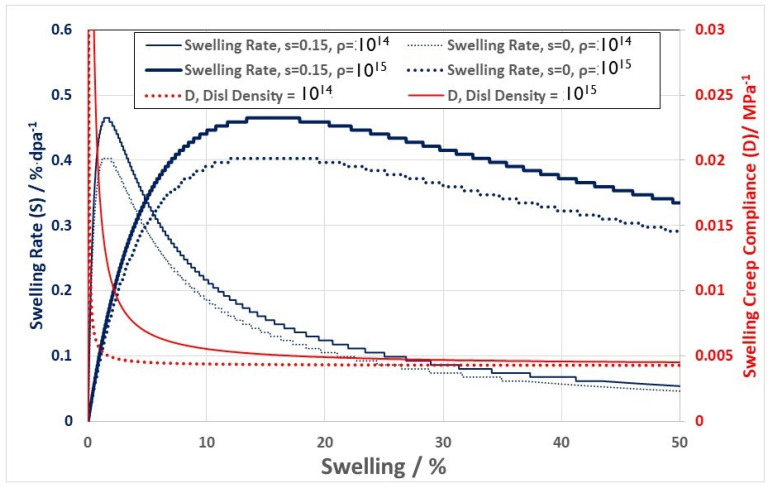
Swelling creep compliance and swelling as a function of swelling for dislocation densities of 10^15^ m^−2^ and of 10^14^ m^−2^. The swelling for an interstitial dislocation bias parameter (p) = 0.3 is shown with and without an applied stress bias (s) = 0.15. The magnitudes of the compliance (D) is given assuming that s = 0.15 is equivalent to 100 MPa.

**Figure 14 materials-16-02287-f014:**
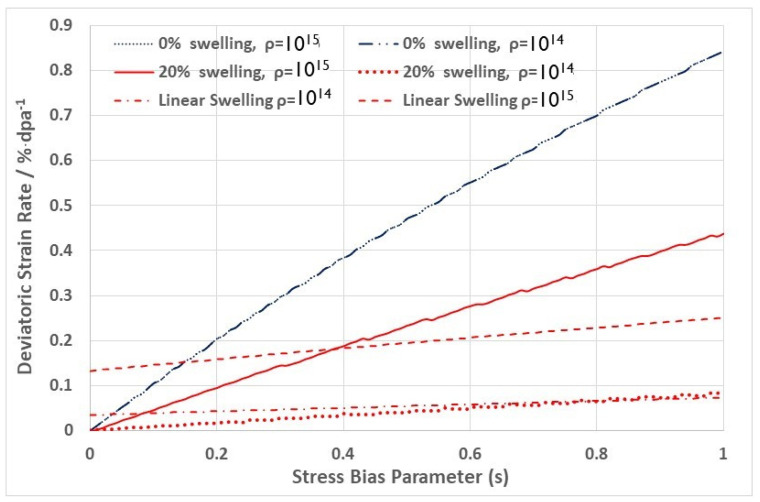
Calculated deviatoric creep and linear swelling strain rates for dislocation densities of 10^15^ m^−2^ and 10^14^ m^−2^ and cavity densities corresponding with 0% and 20% swelling (mean cavity radius = 20 nm).

**Figure 15 materials-16-02287-f015:**
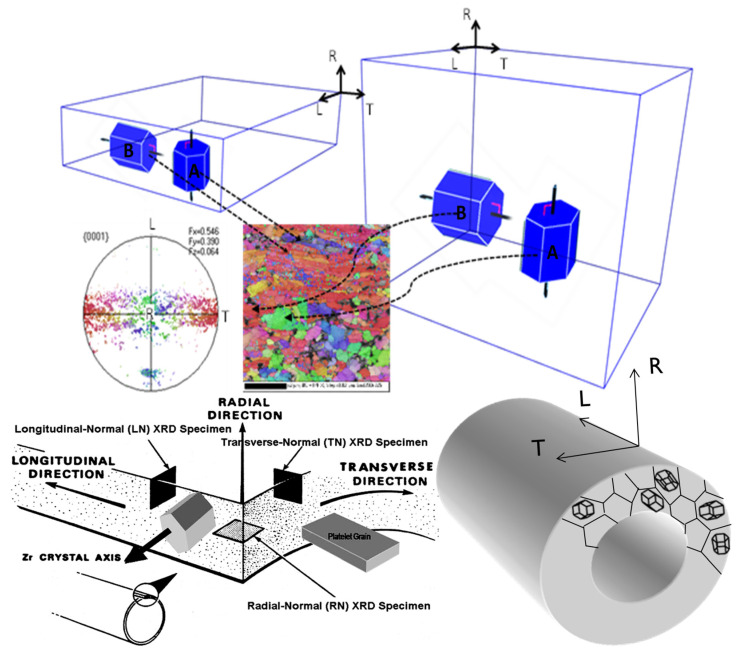
Schematic diagram showing the possible combinations of grain shapes and crystallographic orientations in a cold-worked Zr-2.5Nb pressure tube. Reprinted/adapted with permission from [10]. 2019, Elsevier.

**Figure 16 materials-16-02287-f016:**
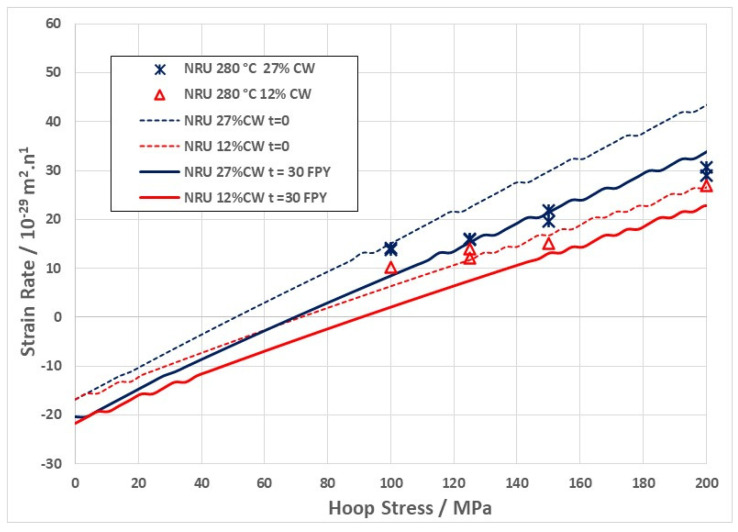
Effect of hoop stress on the diametral strain rate of the 27% CW and 12% CW cases compared with creep capsule data from irradiations in the NRU reactor at 280 °C.

**Figure 17 materials-16-02287-f017:**
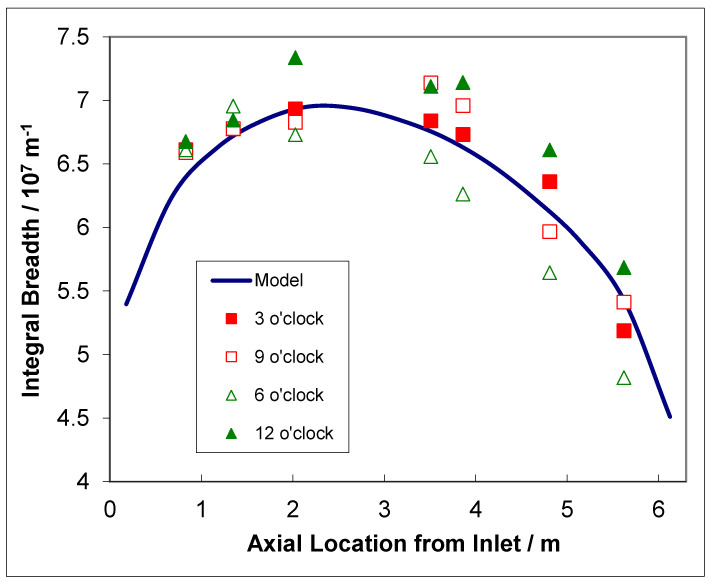
Measured line broadening from a CANDU pressure tube with a symmetric flux profile showing the effect of temperature increase in going from the inlet towards the outlet of the fuel channel. The temperature effect is also seen in the circumferential profile, showing that the bottom of the tube operates at a hotter temperature (especially near the outlet) because the tube expands during operation from diametral creep [46,47].

**Figure 18 materials-16-02287-f018:**
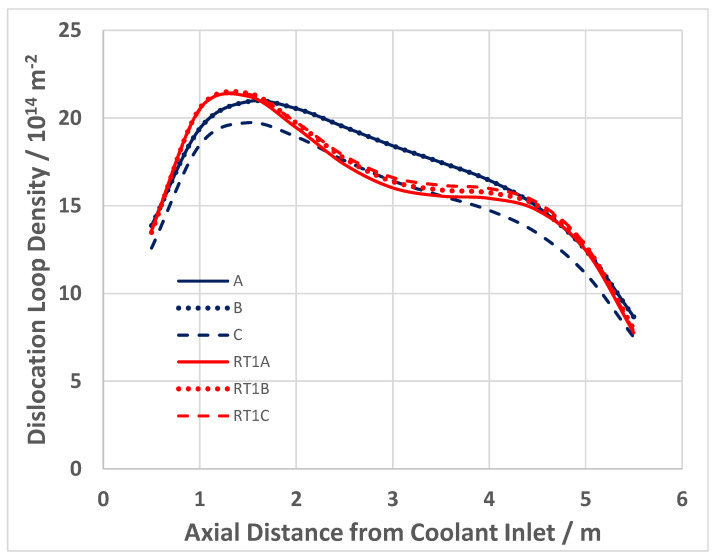
Hypothetical dislocation loop density profiles for different pressure tube operating conditions as a function of axial location.

**Figure 19 materials-16-02287-f019:**
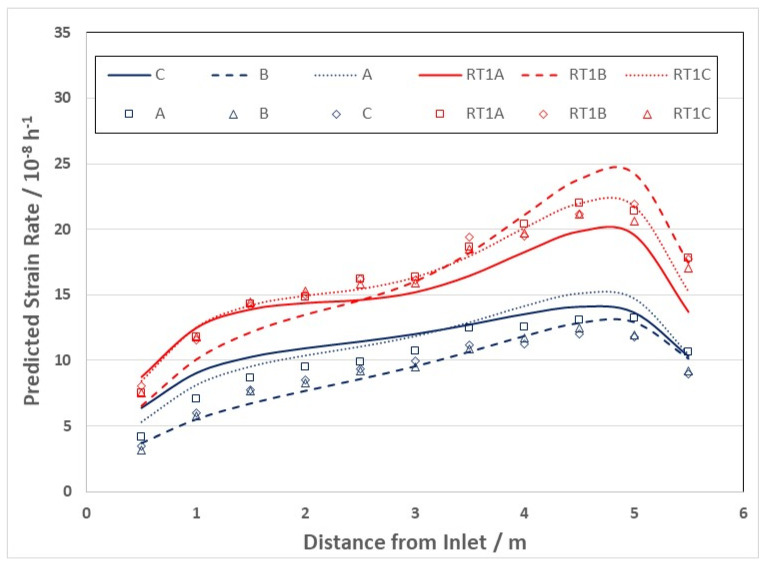
Measured and predicted (assuming a constant hoop stress = 125 MPa) diametral strain rates for a selection of pressure tubes in a CANDU reactor using s = 0.35 and c = 0.2.

**Figure 20 materials-16-02287-f020:**
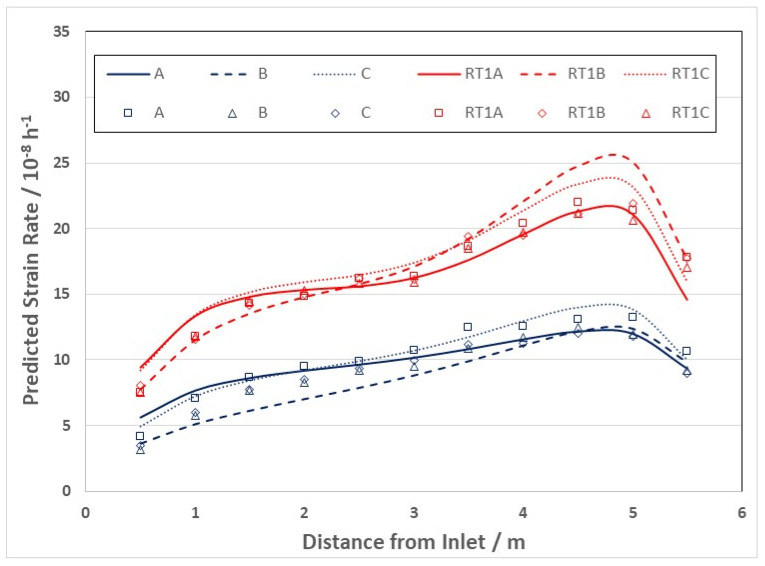
Measured and predicted (assuming a constant hoop stress = 125 MPa) diametral strain rates for a selection of pressure tubes in a CANDU reactor using s = 0.29 and c = 0.04.

**Figure 21 materials-16-02287-f021:**
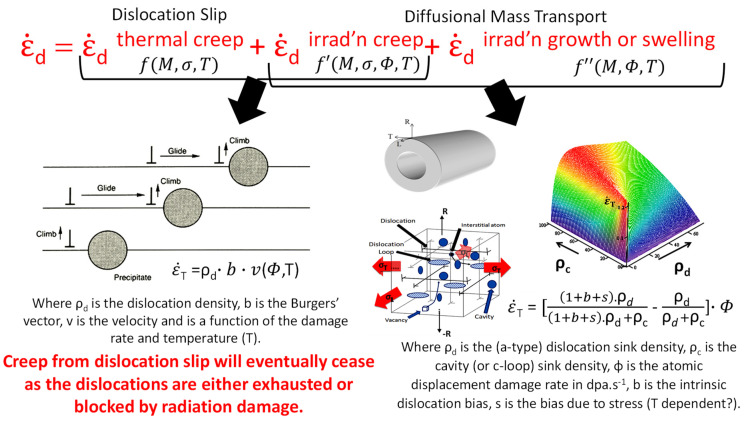
Schematic illustrating the two main contributions to diametral creep for a given stress involving dislocations: (i) that due to dislocation slip; and (ii) that due to diffusional mass transport. The strain is a function of the material (M), the stress (σ), the temperature (T), and the neutron flux or the radiation damage rate (Φ).

**Figure 22 materials-16-02287-f022:**
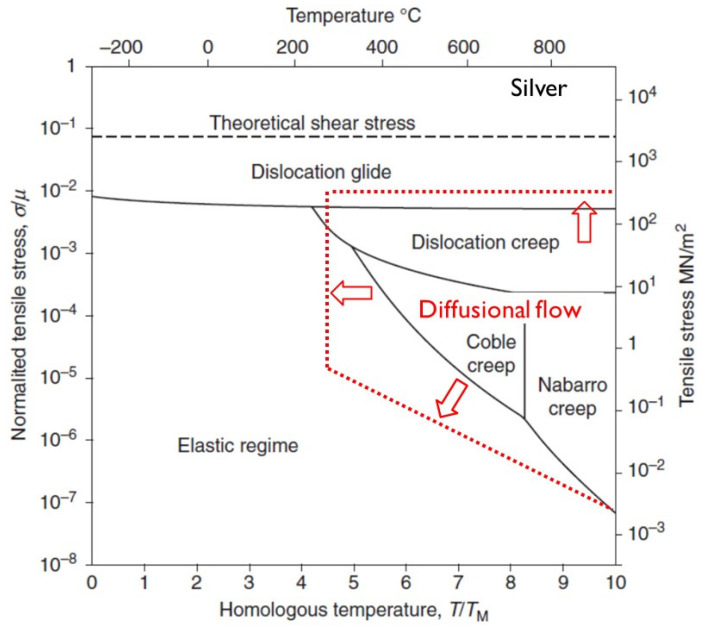
Hypothetical deformation mechanism map for pure silver. The expansion of Nabarro creep shown by the boundaries marked in red indicate the effect of irradiation on the temperature and stress range where bulk diffusional flow is dominant. Reprinted/adapted with permission from [58]. 1972, Elsevier.

**Table 1 materials-16-02287-t001:** Average grain dimensions (minimum grain thickness and aspect ratio looking down the longitudinal axis of the tube), Kearns texture parameters [73] for each tube axis (f_d_), and dislocation densities (ρ_a_ and ρ_c_) for selected tubes installed in a CANDU reactor.

Tube	Location	f_R_	f_T_	f_L_	ρ_a_	ρ_c_	Minor Axis	Aspect Ratio
Designation					10^14^ m^−2^	10^14^ m^−2^	μm	major/minor
A	Front	0.30	0.65	0.05		0.56	0.38	2.61
B	Front	0.28	0.67	0.05	2.12	0.45	0.52	1.91
C	Front	0.29	0.67	0.04	2.05	0.89	0.39	2.57
RT1A	Front	0.40	0.56	0.04	1.53	0.20	0.18	5.57
RT1B	Front	0.38	0.57	0.05	1.35	0.17	0.25	4.07
RT1C	Front	0.38	0.57	0.05	1.38	0.18	0.24	4.14
A	Back	0.33	0.63	0.04	3.84	0.70	0.22	4.46
B	Back	0.34	0.62	0.05	2.39	0.48	0.26	3.83
C	Back	0.31	0.66	0.03	2.38	0.75	0.28	3.53
RT1A	Back	0.38	0.57	0.05	1.38	0.17	0.24	4.24
RT1B	Back	0.41	0.55	0.05	1.40	0.18	0.23	4.41
RT1C	Back	0.40	0.56	0.04	1.48	0.18	0.23	4.35

**Table 2 materials-16-02287-t002:** Measured and predicted (assuming a constant hoop stress = 125 MPa) axial strain rates for a selection of pressure tubes in a CANDU reactor using s = 0.35 and c = 0.2.

Tube	A	B	C	RT1C	RT1B	RT1A
Predicted (P)	10.12	9.66	9.86	5.86	5.82	5.78
Measured (M)	10.06	11.18	9.51	5.56	6.08	5.76
M/P-1	−0.01	0.16	−0.04	−0.05	0.04	0.00

**Table 3 materials-16-02287-t003:** Measured and predicted (assuming a constant hoop stress = 125 MPa) axial strain rates for a selection of pressure tubes in a CANDU reactor using s = 0.29 and c = 0.04.

Tube	A	B	C	RT1C	RT1B	RT1A
Predicted (P)	8.94	10.03	8.75	7.08	7.16	7.14
Measured (M)	10.06	11.18	9.51	5.56	6.08	5.76
M/P-1	0.12	0.11	0.09	−0.21	−0.15	−0.19

## Data Availability

Not applicable.

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
