# Peer review of "Microstructural Effects on Irradiation Creep of Reactor Core Materials"

_materials, 2023, doi:10.3390/ma16062287_

Round 1
Reviewer 1 Report
This manuscript addresses modeling of irradiation creep of Zr-Nb alloy applied in CANDU reactor environment. Indeed, the manuscript is very well written, with extensive description of both experimental and modeling efforts. The authors provided very detailed review of existing literatures on irradiation creep in general and related irradiation creep behaviors to microstructural features, which is of great importance. Irradiation creep has been an area that commonly applied empirical or semi-empirical correlations and models. Fully mechanistic modeling is quite lacking, especially those that relate to detailed microstructural levels.
In my view, this manuscript is directly acceptable to the materials journal. The authors are recommended to check the format of equations as some equal signs are missing, making the equations very odd. In addition, some more details on the rate theory equations may be added.
Author Response
This manuscript addresses modeling of irradiation creep of Zr-Nb alloy applied in CANDU reactor environment. Indeed, the manuscript is very well written, with extensive description of both experimental and modeling efforts. The authors provided very detailed review of existing literatures on irradiation creep in general and related irradiation creep behaviors to microstructural features, which is of great importance. Irradiation creep has been an area that commonly applied empirical or semi-empirical correlations and models. Fully mechanistic modeling is quite lacking, especially those that relate to detailed microstructural levels.
In my view, this manuscript is directly acceptable to the materials journal. The authors are recommended to check the format of equations as some equal signs are missing, making the equations very odd. In addition, some more details on the rate theory equations may be added.
Response: Thank you for the kind comments. I realise that some detailed background on rate theory is warranted but at the same time some detailed background on creep by dislocation slip is also warranted for completeness. To avoid opening up the main text with a detailed explanation of creep mechanisms I have thus created a supplementary file (attached) that describes the mechanistic basis for the two main creep processes (dislocation glide and mass transport using rate theory). I hope that is acceptable.

Reviewer 2 Report
1. Citation numbers are chaotic. Therefore, the list of references should be prepared in the order of citation.
2. There is a noticeable lack of due diligence in presenting theoretical formulas.
3. There is an error in equation (6)—both eq. (6) and (7) are the second invariant from the stress deviator. The expression 1/2 is missing.
4.rr.292 "...k is constants". K=Re/x is a quantity describing the plasticization surface, so it is a material property.
5. Fig. 12 the descriptions in the drawing are difficult to read.
Thanks.
Author Response
- Citation numbers are chaotic. Therefore, the list of references should be prepared in the order of citation.
Response: Apologies - I notice that a link embedded at the end of one reference was assigned a reference # by word when saving and transferring the final document. That put the rest of the references out of sequence – care has been taken to ensure citations are in order.
- There is a noticeable lack of due diligence in presenting theoretical formulas.
Response: Apologies – see 3 and 4.
- There is an error in equation (6)—both eq. (6) and (7) are the second invariant from the stress deviator. The expression 1/2 is missing.
Response: Thank you for pointing out the error. The equations have been corrected.
4.rr.292 "...k is constants". K=Re/x is a quantity describing the plasticization surface, so it is a material property.
Response: Thank you for pointing out the error. The intent was to say that k was a constant for a given material but it will also change as the material deforms and the yield surface expands. The text has been edited.…
“….where k is a value that defines the magnitude of the elastic strain energy needed to plastically deform a given material.”
- Fig. 12 the descriptions in the drawing are difficult to read.
Response: Thank you for pointing out the problem with Figure 12. A revised figure has been created to improve the quality of the hard-to-read text.
Reviewer 3 Report
This is a solid research work and it is worth being published subject to the minor updating/revising:
1. Abstract, vague in results, particularly the microstructural effect
2. Introduction: too elementary and too long, could be much concise;
3. You have referenced some of your publications in introduction, ensure that any published result/diagram is clearly referenced if it has been used in this paper, and also state clearly the similarity and difference in comparison with your existing publications;
4. The numbering of section 2 is wrong;
5. proof read
Author Response
This is a solid research work and it is worth being published subject to the minor updating/revising:
- Abstract, vague in results, particularly the microstructural effect
Response: Thank you for your advice. Abstract has been revised to indicate that, based on the analyses performed, grain structure is deemed an important microstructural parameter controlling irradiation creep.
- Introduction: too elementary and too long, could be much concise;
Response: Thank you for that advice. The introduction has been revised to be more succinct. Two superfluous paragraphs have been removed.
- You have referenced some of your publications in introduction, ensure that any published result/diagram is clearly referenced if it has been used in this paper, and also state clearly the similarity and difference in comparison with your existing publications;
Response: Thank you for that advice. Yes, some figures are similar but not the same as in other publications. They are, however, figures that I have produced and there is no issue with copyright. Also, I reference sources of published data and have made figures based on those data. The figures that have been included are there to help the reader. Appropriate citations have been added where necessary.
- The numbering of section 2 is wrong;
Response: Could not see a problem in my copy – perhaps a glitch in downloading. Headings will be correctly listed as part of the copy edit process.
- proof read
Response: Noted and done – thank you.
Reviewer 4 Report
This manuscript elaborated irradiation creep of zirconium alloy and austenitic steel, especially the influence of microstructure on irradiation creep strain and strain rate. Based on the empirical model, the authors summarized the influence of temperature, stress state, dose rate, and other factors on dislocation loops, dislocation densities, cavity swelling and grain boundary structures, which are all defects promoting or inhibiting irradiation creep. Irradiation creep is a relatively complex process. This review is comprehensive, experienced and clear to be of great significance for publication.
In consideration of improving the quality of this review and its later impact, the importance of irradiation creep on the deformation of reactor structural materials should be stressed in the introduction.
Are there important scientific questions of irradiation creep that remain unanswered? It is better to add some suggestions and prospects for this field at the end of the review to provide greater guidance and help for later scholars.
Author Response
This manuscript elaborated irradiation creep of zirconium alloy and austenitic steel, especially the influence of microstructure on irradiation creep strain and strain rate. Based on the empirical model, the authors summarized the influence of temperature, stress state, dose rate, and other factors on dislocation loops, dislocation densities, cavity swelling and grain boundary structures, which are all defects promoting or inhibiting irradiation creep. Irradiation creep is a relatively complex process. This review is comprehensive, experienced and clear to be of great significance for publication.
Response: Thank you for your kind comments.
In consideration of improving the quality of this review and its later impact, the importance of irradiation creep on the deformation of reactor structural materials should be stressed in the introduction.
Response: The introduction has been revised to stress the importance of irradiation creep. The following has been added….
“Irradiation creep is important when considering: (a) the interaction of fuel with fuel cladding, (b) the limits to operation caused by diametral expansion in CANDU and RBMK reactors and (c) the relaxation of spring components in reactor cores. To this end irradiation creep is largely detrimental to reactor operation. However, stress relaxation by irradiation creep can be beneficial in cases…….”
Are there important scientific questions of irradiation creep that remain unanswered? It is better to add some suggestions and prospects for this field at the end of the review to provide greater guidance and help for later scholars.
Response: Yes, the biggest conundrum is in deciding what proportions of strain can be attributed to dislocation glide and diffusional mass transport. Whereas circumstantial evidence has been presented to indicate which mechanism dominates at what temperature, stress and dose range, a few sentences on potential future work have been added.